# Visual Sparse Steering (VS2): Unsupervised Adaptation for Image Classification via Sparsity-Guided Steering Vectors

## Abstract

Steering vision foundation models at test time, without retraining or access to large labeled datasets, is a desirable yet challenging goal, particularly in dynamic or resource-constrained settings. We present Visual Sparse Steering (VS2), a lightweight, label-free test-time method that constructs a steering vector from sparse features extracted by a Sparse Autoencoder (SAE) trained on the model's internal activations. On CIFAR-100, CUB-200, and Tiny-ImageNet, VS2 improves the top-1 accuracy of the CLIP zero-shot baseline by 4.12%, 1.08%, and 1.84%, respectively. Since not all features learned by the SAE are equally important for classification, we introduce VS2$^{++}$, a retrieval-augmented variant that selectively amplifies relevant sparse features using pseudo-labeled neighbors retrieved from an external unlabeled corpus at inference time. With oracle positive and negative sets (upper bound), VS2$^{++}$ achieves absolute top-1 gains over the CLIP zero-shot baseline of up to 21.44% on CIFAR-100, 7.08% on CUB-200, and 20.47% on Tiny-ImageNet, highlighting the potential of steering vectors when relevant feature selection is accurate. VS2 and VS2$^{++}$ also improve per-class accuracy by up to 25% and 38%, respectively, indicating that sparse steering disproportionately benefits visually or semantically similar classes. Finally, VS2 includes a built-in reliability diagnostic based on SAE reconstruction loss, which is absent in common steering vectors, signaling when steering may underperform and safely triggering a fallback to the baseline.

## 1 Introduction

Foundation Models (FMs) have exhibited strong generalization capabilities across a wide range of domains, from classification tasks to open-ended generation (Radford et al., 2021a; Kirillov et al., 2023; Touvron et al., 2023; Liu et al., 2023; Achiam et al., 2023). Despite their impressive performance, these models largely function as black boxes, limiting their controllability and reliability in practice. Controlling a model's behavior, i.e., editing its outputs, typically involves collecting a labeled dataset that reflects the target objective, followed by supervised fine-tuning (SFT) over the full (Zhang et al., 2023; Dong et al., 2023) or partial (Lester et al., 2021; Hu et al., 2022; Liu et al., 2022) parameter space. However, this process can become prohibitively costly and might lead to catastrophic forgetting. As an alternative, Steering Vector (SV) methods have gained popularity in the Large Language Model (LLM) domain as they can effectively modulate the model's generation at inference time in a training-free manner (Turner et al., 2023; Zou et al., 2023; Liu et al., 2024). However, their use in the vision domain remains underexplored.

Conventional Steering Vectors require positive and negative examples that reflect the behavior we aim to modify. A steering vector is then formed as a directional offset in the latent space, computed by taking the difference between the average latent representations of the contrastive data. This vector is then linearly combined with the latent embedding of a test input to influence the model's behavior. Steering vectors heavily rely on the quality of the contrastive data, whose selection is non-trivial. *Is it though possible to construct effective steering vectors without requiring explicit positive and negative examples?* Additionally, unlike symbolic language tokens, visual representations exhibit high redundancy and entanglement. Consequently, in the vision domain, it is often unclear which features should be steered and in which direction. We thus pose a second question: *How can we*

(a) VS2: Visual Sparse Steering without contrastive data.

(b) VS2$^{++}$: Selective Visual Sparse Steering.

Figure 1: **Overview of VS2 and VS2$^{++}$.** At inference time, VS2 constructs a steering vector by equally amplifying sparse features, while VS2$^{++}$ selectively enhances them when an external visual corpus is available.

*overcome the increased redundancy and entanglement of visual representations to effectively apply steering vectors in the vision domain?*

Drawing inspiration from the mechanistic interpretability literature (Hugofry, 2024; Daujotas, 2024; Bhalla et al., 2024; Stevens et al., 2025), we hypothesize that the sparse features in an SAE's latent space can capture salient aspects of the input and therefore reduce redundancy in the visual representation. SAEs have shown promise in extracting meaningful features from trained foundation models, and are unsupervised, i.e., they do not need positive and negative sets to extract features. Therefore, we ask the question: *Can SAE representations be used effectively for steering?* To this end, we propose **Visual Sparse Steering (VS2)**, a method that constructs steering vectors without requiring labeled anchor examples, by amplifying sparse, non-redundant features extracted from a Sparse Autoencoder (SAE). Across all datasets and ViT backbones, VS2 consistently outperforms the zero-shot CLIP baseline (Radford et al., 2021b) by 3.45% and 4.12% on CIFAR-100, 0.93% and 1.08% on CUB-200, and 1.50% and 1.84% on Tiny-ImageNet with ViT-B/32 and ViT-B/16 backbones, respectively, without relying on external data or supervision, as in common steering vector methods. These results suggest that the sparse features learned through the autoencoding objective can provide a meaningful basis for improving downstream classification performance, as they capture salient and, to some extent, task-relevant information.

We then pose the following question: *Should all selected features be amplified equally, or is there a way to suppress or amplify them non-uniformly?* In vision models, there are potentially many redundant features, e.g., when predicting labels, we want to ignore background features such as the sky. To this end, we relax the problem constraints, allowing the existence of an external vision corpus, and propose **VS2$^{++}$** that leverages this additional unlabeled data. For a given target image, VS2$^{++}$ retrieves the Top-N most similar visual embeddings, and constructs positive and negative groups according to generated pseudo labels. The VS2$^{++}$ Steering Vector is then constructed via subtracting the average SAE features of the negative group from the positive group. Intuitively, VS2$^{++}$ upweights the SAE features most relevant for distinguishing the positive and negative groups. VS2$^{++}$ achieves absolute top-1 accuracy improvements over zero-shot CLIP of up to 21.44% on CIFAR-100, 7.08% on CUB-200, and 20.47% on Tiny-ImageNet when oracle positive and negative sets are available. These results represent an upper bound and highlight the significant headroom for improving steering vectors through more accurate feature selection. When those sets are noisy (non-oracle data), VS2$^{++}$ accuracy drops but still surpasses the standard Steering Vector in most cases, highlighting the need for better feature selection under weakly supervised retrieval.

While a foundational SAE that faithfully captures the data distribution is desirable and remains an active area of research, in practice, test-time inputs may fall outside the training distribution, leading to poor reconstruction quality. In such cases, the sparse features extracted by the SAE may be unreliable, and the resulting steering vectors can degrade model performance. Conventional steering methods do not include mechanisms to determine when steering should or should not be applied, which is a significant limitation especially in high-stakes settings. This raises a critical question: *Can we avoid applying steering when it is likely to hurt performance?* We show that **VS2 includes a built-in reliability diagnostic, a feature absent in prior steering vector methods**. Specifically, when a test input exhibits a high SAE reconstruction error, this provides a signal that the input may be out-of-distribution and that steering may be harmful. In such cases, VS2 falls back to the original zero-shot CLIP prediction, avoiding negative adaptation from misaligned sparse features.

This work investigates *how to control vision foundation models using sparse steering vectors obtained from SAEs* in an unsupervised way. While SAEs have shown promise in mechanistic interpretability, their *effective* application to downstream tasks remains limited. In summary,

- We introduce Visual Sparse Steering (VS2), a lightweight, label-free test-time method that guides vision models using steering vectors derived from sparse features learned by top-k Sparse Autoencoders without requiring contrastive data. VS2 exhibits that steering with as few as 64 sparse activations ($\approx 0.00004\%$ of parameters) yields consistent gains across all datasets, models, and configurations for image classification.

- VS2 also offers a built-in reliability diagnostic, absent from standard steering-vector methods; SAE reconstruction loss can be used at test-time to anticipate when steering may underperform and safely revert to a baseline.

- When an additional external vision corpus is available, VS2$^{++}$ can be used to selectively and robustly amplify important features. Our results highlight the substantial headroom for improving steering vectors through more accurate feature selection and amplification.

## 2 RELATED WORK

**Mechanistic Interpretability and Sparse Autoencoders.** Several traditional approaches exist for interpretability in vision models, including feature visualization (Simonyan et al., 2014; Zeiler & Fergus, 2014; Olah et al., 2017) and network dissection (Bau et al., 2017; Oikarinen & Weng, 2022). Mechanistic interpretability seeks to systematically analyze and understand neural networks (Elhage et al., 2021; Olah et al., 2020), but it faces challenges due to polysemantic neurons i.e. units that activate in response to multiple, seemingly unrelated inputs (Elhage et al., 2022). This phenomenon arises from superposition, where networks encode more features than the available dimensions allow, forcing different concepts to share the same activations (Elhage et al., 2022). Sparse Autoencoders (SAEs) have been explored to mitigate superposition by applying sparse dictionary learning to model internals (Sharkey et al., 2022; Bricken et al., 2023). Recent efforts have leveraged SAEs to uncover interpretable features within LLMs, revealing latent units tied to grammar rules, style patterns, and factual knowledge (Templeton, 2024; Cunningham et al., 2023; Gao et al., 2024a). Joshi et al. (2025) refines this by training SAEs on embedding differences to disentangle multiple concept shifts, enabling precise interventions in model activations without requiring direct supervision. While these methods focus on language models, our work extends sparse steering to the vision domain, where applications of SAEs to **vision models** remain *comparatively* underexplored.

While SAEs have been explored for feature analysis, generative modeling, and concept disentanglement in the visual domain (Hugofry, 2024; Daujotas, 2024; Bhalla et al., 2024; Stevens et al., 2025; Fel et al., 2025; Thasarathan et al., 2025), these works focus primarily on interpretability rather than downstream performance. In contrast, our method leverages SAEs to *actively steer* vision models in a label-free, test-time setting to improve classification. Related efforts like Joseph et al. (2025) study CLIP's steerability on typographic attacks, while Patch-SAE (Lim et al., 2024) improves classification accuracy via *class-conditioned* latent masking. Our approach, on the other hand, requires no class labels and avoids class-based activation aggregation during training, enabling broader applicability without reliance on external supervision or gradient updates.

**Steering vectors.** Steering Vector (SV) methods (Turner et al., 2023; Park et al., 2023; Hernandez et al., 2024; Mikolov et al., 2013), also known as representation engineering (Zou et al., 2023), construct a directional task vector and apply it in the latent space to change the target model's behavior at inference time. In LLMs/MLLMs, SVs are used to enhance security (Liu et al., 2024), truthfulness (Li et al., 2023), reduce hallucinations (Li et al., 2025a), and improve efficiency (Li et al., 2025b). Interestingly, prior work has shown that in VLMs, *visual cues are influenced by language*, and that biases in the model's response can be steered through simple natural language prompts (Gavrikov et al., 2025). In contrast, we focus on steering latent representations in vision models without any language input. Recent work has demonstrated that sparse representations can improve interpretability and disentanglement in steering directions (Bayat et al., 2025; Makelov, 2024). Unlike these methods, which rely on supervised contrastive examples or training data, our approach discovers meaningful sparse directions in vision models without requiring labeled positive/negative concept pairs, making it more adaptable to general visual representations.

**Test-Time Adaptation.** Recent advances in improving the generalization of vision–language models have introduced a variety of prompt-tuning approaches, such as CoOp (Zhou et al., 2022b), CoCoOp (Zhou et al., 2022a), and MaPLe (Khattak et al., 2023), along with adapter-based techniques like Tip-Adapter (Zhang et al., 2021) and CLIP-Adapter (Gao et al., 2024b). Despite their effectiveness, these methods typically assume access to a small number of labeled target-domain examples (often in a few-shot setting). In contrast, Test-Time Adaptation methods aim to improve robustness under distribution shift by adapting a pre-trained model using only unlabeled test samples at inference time (Liu et al., 2021; Sun et al., 2020; Wang et al., 2020; Gao et al., 2022). Specifically, Test-Time Prompt Tuning (TPT) (Shu et al., 2022) introduced entropy-minimizing prompt optimization over augmented views, inspiring subsequent methods such as DiffTPT (Feng et al., 2023), which leverages diffusion-based augmentations, and C-TPT (Yoon et al., 2024), which incorporates calibration-aware objectives. Although effective, these strategies typically require optimization loops, multiple augmentations, and backpropagation through large encoders, resulting in substantial computational and memory overhead at inference. In contrast, VS2 performs a single forward pass with sparse SAE-guided steering, avoiding test-time optimization while still providing measurable gains.

## 3 METHOD

**Preliminaries.** A Sparse Autoencoder (SAE) consists of an encoder-decoder pair that maps an input vector $x$ into a latent representation $z$, then reconstructs $x$ from $z$. Formally, we define the encoder as: $z = \text{ReLU}(W_{\text{enc}}(x - b_{\text{pre}}) + b_{\text{enc}})$, where $W_{\text{enc}} \in \mathbb{R}^{n \times d}$ is the encoder weight matrix, $b_{\text{enc}} \in \mathbb{R}^n$ is a bias term, and $b_{\text{pre}} \in \mathbb{R}^d$ normalizes the input. The decoder then reconstructs $x$ using a decoder weight matrix , as $\hat{x} = W_{\text{dec}} z + b_{\text{pre}}$. To enforce sparsity, $\ell_1$ norm with strength $\alpha$ is introduced as the regularization term, and the intact optimization objective becomes $\mathcal{L}_{\text{SAE}} = \|x - \hat{x}\|_2^2 + \alpha \|z\|_1$. In practice, instead of relying on $\ell_1$ regularization, top-$k$ Sparse Autoencoder (Gao et al., 2024a) explicitly enforces the sparsity by using a **top-$k$ selection** mechanism: $z = \text{TopK}(W_{\text{enc}}(x - b_{\text{pre}}))$, where $\text{TopK}(\cdot)$ operator retains only the $k$ highest-magnitude activations and zeroes out the rest. This modification ensures that only a fixed number of latent dimensions contribute to the reconstructed representation, making the extracted features more interpretable.

### 3.1 VISUAL SPARSE STEERING

**Visual Sparse Steering (VS2)** is a lightweight, label-free, test-time method that guides vision models using steering vectors derived from sparse features learned by top-k Sparse Autoencoders without requiring contrastive data. We hypothesize that the sparse features in an SAE's latent space can capture salient aspects of the input, thereby reducing redundancy. To this end, VS2 constructs a steering vector by amplifying the SAE's salient features, reconstructing the corresponding amplified representation, and subtracting the original reconstruction (without amplification) to minimize the inherent SAE's reconstruction errors. The resulting sparse steering vector points in the direction of the salient features and is linearly combined with the original embedding to steer the model's behavior. *Unlike common steering vectors, VS2 does not require positive and negative anchor examples*; instead, steering occurs in the direction of the most salient features identified by the SAE.

#### 3.1.1 VS2: VISUAL SPARSE STEERING

**Learning latent concept space.** Given an input image $X \in \mathcal{X}$ and its corresponding embedding $x \in \mathbb{R}^d$ extracted from layer $i$ of a Vision Foundation Model (VFM), we define a *Concept Encoder* $E_c \colon \mathbb{R}^d \to \mathbb{R}^{|\mathbb{C}|}$ that maps the embedding $x$ into a sparse feature activation vector: $c = E_c(x) = (c_1, \ldots, c_{|\mathbb{C}|})$. By imposing sparsity constraints described in Sec. **??** on the feature activations, the encoder $E_c$ is encouraged to identify *disentangled*, semantically meaningful features. To map the features into the original representation space, we define a *Concept Decoder* $D_c \colon \mathbb{R}^{|\mathbb{C}|} \to \mathbb{R}^d$ that reconstructs the original embedding from these features: $\tilde{x} = D_c(c)$. Since explicit supervision for feature identification is typically unavailable, we train the encoder-decoder pair jointly on training data $\mathcal{X}$, enforcing both sparsity and accurate embedding reconstruction to implicitly uncover meaningful features.

**Constructing steering vector.** At inference time, emerged sparse features are assumed semantically significant and critical for downstream tasks, and we enhance them via a *Concept Upweighting*

*Function* $U : \mathbb{R}^{|\mathbb{C}|} \to \mathbb{R}^{|\mathbb{C}|}$. In implementation, we simply scale the original feature activations $c$ by a factor of $\gamma$ to amplify their effects: $c' = U(c) = \gamma \times c$. Reconstructing this modified feature vector $c'$ provides a conceptually steered embedding: $\tilde{x}' = D_c(c')$. Since reconstruction steps inherently introduce approximation errors, we define the steering vector $v$ as: $v = \tilde{x}' - \tilde{x}$. which explicitly represents the semantic shift induced by upweighting and aims to mitigate the effects of the reconstruction errors.

**Steer the target emebedding.** Obtaining the steering vector $v$ that aims to amplify the latent features embedded in $x$, we shift the original embedding by adding the steering vector back as: $\hat{x} = x + \lambda v$. where $\lambda$ is a hyperparameter that controls the magnitude of steering. To stabilize the representation, we further rescale the $\ell_2$ norm of the steered embedding to its original scale: $\hat{x} = \frac{\hat{x} \cdot \|x\|_2}{\|\hat{x}\|_2}$. Intuitively, this procedure precisely moves the embedding $x$ along directions defined by semantically relevant, salient features, thus enhancing robustness against spurious correlations. We refer Fig. 1 (left) for a schematic overview of VS2.

### 3.1.2 VS2$^{++}$: Visual Sparse Steering with Unlabeled Data

In VS2, we leverage the learned SAEs to identify the most salient features for a given latent representation, but treat all identified features equally important by equally amplifying them. In vision models, there are potentially many redundant features, e.g., when predicting labels, we want to ignore background features such as the sky. To selectively amplify only important features more robustly and non-uniformly, we relax the problem, allowing the existence of an unlabeled external vision corpus, and propose VS2$^{++}$ that leverages this additional unlabeled data to construct steering vectors. For a given target image, VS2$^{++}$ retrieves the Top-N most similar visual embeddings and constructs positive and negative groups according to generated pseudo labels. The VS2$^{++}$ Steering Vector is then constructed via subtracting the average SAE features of the negative group from the positive group. Intuitively, VS2$^{++}$ amplifies the SAE features most relevant for distinguishing the positive and negative groups.

**Constructing Contrastive Groups.** Given a query embedding $x_q$ and an unlabeled dataset of embeddings $\{x_i\}_{i=1}^M$, VS2$^{++}$ first retrieve the set of $N$ nearest neighbors $\mathcal{N}_N(x_q)$ for a query embedding $x_q$ is defined as:

$$\mathcal{N}_N(x_q) = \underset{\substack{\{x_i\} \subseteq \{x_i\}_{i=1}^M \\ |\mathcal{N}_N(x)| = N}}{\arg\max} S(x_q, x_i), \tag{1}$$

where $S(,)$ denotes the measurement of cosine similarity. After retrieving $\mathcal{N}_N(x_q)$, we obtain the pseudo-label for the query embedding $\hat{y}_q = \arg\max_y P(y \mid x_q)$, where $P(y \mid x)$ is the classifier's predicted probability for label $y$. Similarly, for each neighbor $x_i \in \mathcal{N}_N(x_q)$, we compute its pseudo-label: $\hat{y}_i = \arg\max_y P(y \mid x_i)$. We then construct the *positive* group, $\mathbb{S}^+(x_q)$, by selecting neighbors sharing the same pseudo-label as the query embedding: $\mathbb{S}^+(x_q) = \{x_i \in \mathcal{N}_N(x_q) \mid \hat{y}_i = \hat{y}_q\}$. Likewise, the *negative* group, $\mathbb{S}^-(x_q)$, is formed by the remaining neighbors: $\mathbb{S}^-(x_q) = \mathcal{N}_N(x_q) \setminus \mathbb{S}^+(x_q)$.

**Selective Steering.** For each embedding $x_i \in \mathbb{S}^+(x_q)$, VS2$^{++}$ first obtains the steering vectors as in VS2 acquiring a directional vector $v_i^p$ that underscores the underlying features within the given embedding. We repeat the same procedure for each $x_j \in \mathbb{S}^-(x_q)$ to obtain the negative steering vectors $v_j^n$. We then average these per-embedding steering vectors within each group, yielding the *positive anchor*

$$\bar{v}^p = \frac{1}{|\mathbb{S}^+(x_q)|} \sum_{x_i \in \mathbb{S}^+(x_q)} v_i^p, \tag{2}$$

and the *negative anchor*

$$\bar{v}^n = \frac{1}{|\mathbb{S}^-(x_q)|} \sum_{x_j \in \mathbb{S}^-(x_q)} v_j^n. \tag{3}$$

Finally, we obtain the *contrastive steering vector* by subtracting the negative anchor from the positive anchor: $v = \bar{v}^p - \bar{v}^n$, and apply it to the query embedding $x_q$ in the same manner as VS2:

$$\hat{x}_q = x_q + \lambda v, \quad \hat{x}_q = \frac{\hat{x}_q \cdot \|x_q\|_2}{\|\hat{x}_q\|_2}. \tag{4}$$

Table 1: Benchmarking Visual Sparse Steering: Zero-Shot Accuracy (%) with and without External Data on CIFAR-100, CUB-200, and Tiny Imagenet using ViT-B/32 and ViT-B/16.

| Method | CIFAR-100 | | CUB-200 | | Tiny-IN | |
|---|---|---|---|---|---|---|
| | ViT-B/32 | ViT-B/16 | ViT-B/32 | ViT-B/16 | ViT-B/32 | ViT-B/16 |
| **Label-free, unsupervised, test-time steering** | | | | | | |
| $CLIP_{ZS}$ | 61.07 ( 0 ) | 63.96 ( 0 ) | 51.76 ( 0 ) | 55.06 ( 0 ) | 56.64 ( 0 ) | 61.08 ( 0 ) |
| $SAE_{REC}^{A}$ | 58.01 ( −3.06 ) | 64.05 ( +0.09 ) | 47.45 ( -4.31 ) | 51.81 ( -3.25 ) | 30.56 ( −26.08 ) | 52.96 ( -8.12 ) |
| $SAE_{REC}^{F}$ | 58.22 ( −2.85 ) | 63.42 ( −0.54 ) | 48.08 ( -3.68 ) | 51.43 ( -3.63 ) | 36.33 ( −20.31 ) | 54.84 ( -6.24 ) |
| $SAE_{REC}^{F+\gamma}$ | 62.69 ( +1.62 ) | 66.81 ( +2.85 ) | 49.41 ( -2.35 ) | 53.28 ( -1.78 ) | 39.49 ( −17.15 ) | 58.81 ( -2.27 ) |
| **VS2 (ours)** | **64.52** ( +3.45 ) | **68.08** ( +4.12 ) | **52.69** ( +0.93 ) | **56.14** ( +1.08 ) | **58.14** ( +1.50 ) | **62.92** ( +1.84 ) |
| **RAG-enhanced (oracle unlabeled external data)** | | | | | | |
| Weighted RAG | 76.43 ( +15.36 ) | 69.78 ( +5.82 ) | 58.32 ( +6.56 ) | 60.65 ( +5.59 ) | 72.53 ( +15.89 ) | 67.84 ( +6.75 ) |
| CLIP Steering Vector | 81.85 ( +20.78 ) | 84.12 ( +20.16 ) | 56.42 ( +4.66 ) | 61.51 ( +6.45 ) | **80.38** ( +23.74 ) | 84.07 ( +22.99 ) |
| **VS2++** | **81.95** ( +20.88 ) | **85.40** ( +21.44 ) | **58.84** ( +7.08 ) | **61.91** ( +6.85 ) | 73.27 ( +16.63 ) | **81.55** ( +20.47 ) |
| **RAG-enhanced (non-oracle unlabeled external data)** | | | | | | |
| CLIP Steering Vector | 77.22 ( +16.15 ) | 78.97 ( +15.01 ) | 47.89 ( -3.87 ) | 53.95 ( -1.11 ) | **74.06** ( +17.42 ) | 76.79 ( +15.71 ) |
| **VS2++** | 77.11 ( +16.04 ) | **79.14** ( +15.18 ) | **52.81** ( +1.05 ) | **57.02** ( +1.96 ) | 72.12 ( +15.48 ) | **76.92** ( +15.84 ) |

This selective scheme thus anchors the steering directions in features consistently activated within the positive group, while suppressing undesired features highlighted by the negative group. Since $VS2^{++}$ retrieves N-nearest embeddings of a query to construct both positive and negative groups, negative group contains non-trivial hard cases. We refer Fig. 1 (right) for a schematic overview of $VS2^{++}$.

## 4 EXPERIMENTS

We evaluate VS2 and $VS2^{++}$ on three datasets: CIFAR-100 (Krizhevsky et al., 2009), Tiny-ImageNet (Le & Yang, 2015), and CUB-200 (Wah et al., 2011), covering standard, complex, and fine-grained classification tasks. For the vision foundation model, we use CLIP (Radford et al., 2021b) with ViT-B/32, and ViT-B/16 backbones. Our goal is to investigate the effectiveness of Sparse Autoencoders (SAEs) for constructing steering vectors. In Sections 4.1 and 4.2, we focus on the case where the SAE can faithfully reconstruct the test inputs. To this end, we train top-$k$ SAEs on CLS token embeddings extracted from each layer of CLIP's vision transformer, using the training split of each dataset to learn sparse latent representations (unsupervised in-domain data). Additional training details are provided in Appendix C. In Section 4.4, we relax this assumption by training a single, more general SAE across all train datasets, including both general and fine-grained tasks, to better approximate a real-world deployment scenario and evaluate how VS2 can avoid harmful steering when test samples cannot be faithfully reconstructed. This capability is novel compared to common steering vector techniques.

### 4.1 VS2: VISUAL SPARSE STEERING WITHOUT CONTRASTIVE DATA

**Baselines.** We evaluate the effectiveness of VS2 on downstream classification tasks. As baselines, we report: (1) the zero-shot performance of the original CLIP model ($CLIP_{ZS}$); (2) $CLIP_{REC}^{F}$, which uses the SAE-reconstructed CLS token from the final layer without steering; and (3) $CLIP_{REC}^{A}$, which uses reconstructed CLS tokens from all layers. These isolate the effect of reconstruction alone, with no steering applied. All top-$k$ SAEs are trained on CLS token activations across all transformer layers. We also include $SAE_{REC}^{F+\gamma}$, which applies a fixed scaling ($\gamma = 1.5$) to the top-$k$ sparse features before reconstructing the final-layer CLS token, isolating the impact of latent amplification. Appendix G provides pseudocode comparing these baselines to VS2. Additional comparisons with Splice (Bhalla et al., 2024), which uses an external vocabulary to define latent features (unlike our unsupervised SAEs), are included in Appendix H. VS2 steering is applied to the CLS token at the final layer of the vision encoder. Appendix E provides further analysis of VS2 steering across layers.

**Results.** From Table 1, we demonstrate that using only SAE reconstructions generally reduces zero-shot performance in all datasets; we hypothesize this is due to reconstruction errors in SAEs

Table 2: Accuracy and efficiency comparison of VS2 and baselines.

(a) UDA comparison across datasets.

| Method | CIFAR-100 | CUB-200 | Tiny ImgNet |
|---|---|---|---|
| Zero-shot | 61.07 | 51.76 | 56.64 |
| Ensemble (Radford et al., 2021b) | 63.66 | 51.54 | 61.39 |
| TPT (Shu et al., 2022) | 64.09 | 51.83 | 62.77 |
| C-TPT (Yoon et al., 2024) | 64.86 | 52.54 | **63.20** |
| Diff-TPT (Feng et al., 2023) | 63.04 | **52.80** | 61.60 |
| **VS2 (ours)** | 64.52 | 52.69 | 58.14 |
| VS2 + Ensemble (Radford et al., 2021b) | 65.48 | 52.47 | 62.79 |
| VS2 + Ensemble + Aug | **65.80** | 51.54 | 62.59 |

(b) Inference-time compute (CLIP ViT-B/32).

| Method | GFLOPs | GMACs | Params (M) |
|---|---|---|---|
| Plain CLIP (vision) | 8.7295 | 4.3623 | 87.456 |
| VS2 (SAE steering) | 8.7342 | 4.3647 | 92.178 |
| LoRA (r = 16) | 8.7885 | 4.3918 | 88.046 |

(Engels et al., 2025). However, when SAE features are amplified in the reconstruction, performance can improve, as demonstrated by $\text{SAE}_{\text{REC}}^{\text{F}+\gamma}$ for CIFAR-100. Specifically, $\text{SAE}_{\text{REC}}^{\text{F}+\gamma}$ outperforms the zero-shot baseline on CIFAR-100 by 1.62% and 2.85% using the ViT-B/32 and ViT-B/16 backbones respectively. However, $\text{SAE}_{\text{REC}}^{\text{F}+\gamma}$ is worse than the CLIP baseline on CUB-200 and Tiny-ImageNet. In contrast, **our method VS2 consistently outperforms all baseline methods across all datasets and ViT backbones**, surpassing zero-shot CLIP by 3.45% and 4.12% on CIFAR-100, 0.93% and 1.08% on CUB-200, and 1.50% and 1.84% on Tiny-ImageNet with ViT-B/32 and ViT-B/16 backbones, respectively. **In summary, we demonstrate that our proposed steering vectors using SAEs can improve downstream classification performance on vision tasks without relying on any external data or supervision**. Although our focus is on *how* to effectively steer SAEs, we also provide some evidence in Appendix A that the sparse features appear to capture subtle class-specific attributes and concepts, such as birds with gray upperparts, white underparts, a white-colored throat, and black eyes. Finally, in Appendix B, we present a sensitivity analysis of the VS2 hyperparameters $\gamma$ and $\lambda$, demonstrating the robustness of our proposed methods.

**Comparison with Test-Time Adaptation Baselines.** In Table 2a, we compare our method against Unsupervised Domain Adaptation (UDA) approaches. Specifically, we include baselines from TPT (Shu et al., 2022), C-TPT (Yoon et al., 2024), and Diff-TPT (Feng et al., 2023) using ViT-B/32. These methods optimize text prompts during adaptation, whereas our method does not modify or update any text prompts. The Ensemble setting corresponds to a set of 80 fixed ImageNet-style templates, following CLIP's standard zero-shot evaluation protocol (Radford et al., 2021a). Finally, Aug denotes the test-time image augmentations used in TTA baselines. For efficiency, we apply only 10 augmentations, compared to the 63 used in prior work. We observe that VS2 performs comparably to existing Test-Time Adaptation techniques, achieving the highest accuracy on CIFAR-100 and the second-best performance on CUB-200 and TINY IMAGENET. Notably, a key advantage of our approach lies in its inherent interpretability, an aspect absent from the compared methods, as shown in Appendix A.2. Furthermore, VS2 is modality-agnostic and introduces minimal computational overhead.

**Computational Overhead.** VS2 is designed to be a lightweight alternative to parameter- or prompt-based test-time adaptation. Table 2b reports the inference-time cost of a CLIP ViT-B/32 vision encoder under different adaptation strategies. Adding VS2 increases the vision encoder FLOPs by only 0.0047 GFLOPs (less than 0.1%), since the SAE operates on the [CLS] token at a single layer. In contrast, a LoRA adapter with rank 16 must be applied inside every attention block, leading to noticeably higher FLOPs despite having fewer trainable parameters. At training time, learning the SAE is also inexpensive. With cached activations, SAE training requires only 0.014 GFLOPs per sample, compared to roughly 26 GFLOPs for LoRA or full fine-tuning of the vision encoder. Full training-time FLOPs and parameter counts are reported in Table 17 in Appendix N. Finally, compared to Test-Time Adaptation (TTA) methods such as TPT (Shu et al., 2022), VS2 is orders of magnitude cheaper at inference. A single CLIP forward pass on CIFAR-100 with 100 text prompts costs about 0.59 TFLOPs. In our implementation, TPT performs 63 augmented views and 4 optimization steps with forward and backward passes, leading to approximately 9.8 TFLOPs per test image, around 16.6× the cost of plain CLIP (see Appendix N for details). In contrast, VS2 requires only one additional SAE forward pass and keeps the per-image overhead below 0.01% of CLIP.

## 4.2 VS2$^{++}$: Visual Sparse Steering with Selectively Amplifying Features

In contrast to VS2, which amplifies the salient SAE features equally, VS2$^{++}$ serves as a method to selectively amplify or suppress features non-uniformly. To achieve this, we relax the experimental setup by allowing the use of external data, i.e., data from the training set of each dataset, to guide steering more selectively. As a proxy for identifying salient features, we adopt a Retrieval-Augmented Generation (RAG) approach: we retrieve the top-50 most similar training images to the test input using DINOv2 (Oquab et al., 2023) and average their sparse activations. To construct the positive and negative sets, we assign pseudolabels using a CLIP classifier; images with the most frequent pseudolabel form the positive set, while the remaining images serve as negatives. This setup corresponds to the non-oracle data in Table 1. For comparison, we also report an ideal scenario that serves as an upper-bound, i.e., an oracle setting that uses ground-truth labels of the retrieved images to define positives and negatives. As a baseline, we use the Retrieval-Augmented Generation (RAG) pipeline (Lewis et al., 2020), by combining the test query embedding with a weighted aggregation of retrieved image embeddings. Additional details on the weighting strategy are provided in Appendix D. We also include a non-SAE variant that constructs a steering vector by averaging the CLIP embeddings of the positive and negative sets and taking their difference, yielding a purely contrastive direction without relying on Sparse Autoencoders.

**Results.** In Table 1, we present the results for VS2$^{++}$ when external data is available under both settings, where we have oracle and non-oracle data. When oracle positive/negative sets are available, we observe that VS2$^{++}$ surpasses all other methods under almost every setting. Weighted RAG performs worse than the steering vector approaches (both CLIP and VS2$^{++}$), suggesting that directionality is more informative than proximity. In the more realistic scenario of non-oracle positive and negative sets, we observe drops in accuracy. We hypothesize that this degradation is due to inaccuracies in pseudolabeling, resulting in the amplification of spurious directions in the SAE latent space. Across the six configurations, VS2$^{++}$ outperforms the CLIP Steering Vector in four cases and falls behind only in the ViT-B/32 runs for CIFAR-100 and Tiny-ImageNet (with CIFAR-100 essentially on par). These results suggest that there remains significant room for improvement in actively selecting which sparse features to steer, especially under noisy or weakly supervised retrieval scenarios. In Appendix I, we conduct ablation studies on the top-$N$ retrieved images that indicate that the utility of neighbor-based aggregation is dataset-dependent: general object recognition tasks may benefit from larger $N$, while fine-grained classification may require retrieving fewer samples to avoid introducing noise from visually similar but irrelevant examples.

## 4.3 Fine-Grained Per-Class Accuracy Analysis

To investigate *which classes* Visual Sparse Steering helps most, we compute per-class top-1 accuracy on CIFAR-100 with a ViT-B/32 backbone and report the ten largest improvements in Table 3. The results reveal a long-tailed pattern: **VS2 lifts accuracy by up to 25 % in individual classes** (e.g., *lawn-mower → tractor*, *pine-tree → forest*), while **VS2$^{++}$ reaches gains of 38 %** when high-quality neighbors are available (e.g., *bee → spider*, *lion → tiger*, *whale → flatfish*). In all cases, the corrected predictions involve visually or taxonomically proximate categories, suggesting that sparse steering helps refine decision boundaries between semantically similar classes. This observation is consistent with the qualitative examples shown in Appendix A.2, where the top-$k$ SAE features appear to highlight subtle, class-relevant attributes such as gray upperparts, white underparts, a white-colored throat, and black eyes in bird images. A more detailed list of per-class performance accuracies of VS2 and VS2$^{++}$ is provided in Table 15 in Appendix J.

## 4.4 SAEs as a Reliability Diagnostic for Safe Visual Sparse Steering

Our approach assumes access to a Sparse Autoencoder (SAE) that models the input data distribution. In practice, we approximate this by training SAEs on the in-domain training split of each dataset, which allows us to *isolate and investigate the effectiveness of sparsity-guided steering vectors* in modulating the classification behavior of CLIP's vision transformer.

In practice, constructing a universal, high-fidelity SAE that generalizes across diverse distributions is a challenging and ongoing research problem. As a result, when test-time data is out of the SAE's learned distribution (OOD), the extracted sparse features may be unreliable, potentially leading to

Table 3: **Top-10 class gains on CIFAR-100.** Top-1 accuracy; green = gain over CLIP Zero-shot.

(a) **VS2**

| Class | ZS | VS2 (+Δ) | Mislabel | Why visually confusing |
|---|---|---|---|---|
| tractor | 55.0 | 80.0 ↑(+25.0) | lawn_mower | Both are wheeled agricultural machines; similar viewed side-on. |
| forest | 35.0 | 58.0 ↑(+23.0) | pine_tree | Dense tree canopies often dominated by tall conifers. |
| man | 53.0 | 74.0 ↑(+21.0) | boy | Male human silhouette; age difference subtle at CIFAR resolution. |
| bus | 51.0 | 68.0 ↑(+17.0) | pickup_truck | Rectangular vehicle profile with windows and wheels. |
| snake | 61.0 | 76.0 ↑(+15.0) | worm | Long, legless bodies with smooth textures. |
| woman | 57.0 | 71.0 ↑(+14.0) | girl | Female human figures; height/size cues are small. |
| trout | 18.0 | 32.0 ↑(+14.0) | aquarium_fish | Similar fish shape and reflective scales in water scenes. |
| cockroach | 49.0 | 62.0 ↑(+13.0) | beetle | Small, dark exoskeleton insects with segmented bodies. |
| bridge | 72.0 | 83.0 ↑(+11.0) | sea | Large spans photographed above wide blue water areas. |
| rose | 58.0 | 68.0 ↑(+10.0) | poppy | Red, multi-petal flowers with central dark core. |

(b) **VS2++**

| Class | ZS | VS2++ (+Δ) | Mislabel | Why visually confusing |
|---|---|---|---|---|
| spider | 53.0 | 91.0 ↑(+38.0) | bee | Small multi-legged insects; dark radial silhouettes. |
| tiger | 48.0 | 84.0 ↑(+36.0) | lion | Large orange/brown big cats; stripes vs. mane often unclear at low resolution. |
| flatfish | 49.0 | 85.0 ↑(+36.0) | whale | Flattened marine shapes against blue water backgrounds. |
| possum | 30.0 | 65.0 ↑(+35.0) | hamster | Small brown furry mammals with rounded bodies. |
| lizard | 39.0 | 74.0 ↑(+35.0) | snake | Slender reptiles; legs on lizard hard to spot in 32 × 32 images. |
| sweet_pepper | 58.0 | 92.0 ↑(+34.0) | orange | Round orange-coloured produce with glossy skins. |
| raccoon | 49.0 | 83.0 ↑(+34.0) | skunk | Mid-sized nocturnal mammals with dark fur and contrasting markings. |
| shrew | 46.0 | 79.0 ↑(+33.0) | skunk | Small ground mammals; dark elongated silhouettes blur distinguishing cues. |
| snail | 53.0 | 86.0 ↑(+33.0) | mushroom | Brown dome (shell vs. cap) resting on the forest floor. |
| beetle | 49.0 | 82.0 ↑(+33.0) | cockroach | Dark, shiny exoskeletal insects with similar body shapes. |

Table 4: Out-of-distribution performance of VS2 using an in-domain SAE, a generalized SAE, and a generalized SAE with a reconstruction-loss-based fallback. The fallback threshold $\tau$ controls when steering is applied.

| Method / Dataset | Aircraft | Food101 | Flower102 | Caltech101 | SUN | EuroSAT |
|---|---|---|---|---|---|---|
| Baseline (CLIP$_{ZS}$) | 24.84 | 80.71 | 63.72 | 78.53 | 51.79 | 31.61 |
| VS2 (In-domain SAE) | 27.69 (+2.85) | 81.20 (+0.49) | 64.81 (+1.09) | 80.95 (+2.42) | 54.80 (+3.01) | 34.65 (+3.04) |
| VS2 (Generalized SAE) | 18.57 (-6.27) | 81.06 (+0.35) | 64.11 (+0.39) | 79.72 (+1.19) | 45.78 (-6.01) | 30.76 (-0.85) |
| VS2 + Fallback ($\tau$=2.0) | 24.87 (+0.03) | 80.72 (+0.01) | 63.70 (-0.02) | 78.53 (+0.00) | 51.79 (+0.00) | 31.61 (+0.00) |
| VS2 + Fallback ($\tau$=4.0) | 24.87 (+0.03) | 80.71 (+0.00) | 63.88 (+0.16) | 78.11 (-0.42) | 51.79 (+0.00) | 30.41 (-1.20) |
| VS2 + Fallback ($\tau$=6.5) | 18.69 (-6.15) | 81.05 (+0.34) | 64.11 (+0.39) | 79.72 (+1.19) | 45.86 (-5.93) | 30.76 (-0.85) |

harmful steering. To approximate a more realistic setting, we trained a more generalizable SAE by jointly using the union of the unlabeled training splits of CIFAR-100, CUB-200, and Tiny-ImageNet, instead of training a separate SAE per dataset. We then evaluate this single SAE per dataset, reporting FVU reconstruction and top-1 classification accuracy. The results are presented in Table 5. We observe that, on CIFAR-100 and Tiny-ImageNet, the generalized SAE yields strong accuracy gains over the zero-shot baseline, demonstrating that our approach can remain effective even when the SAE is trained on a more general input distribution. However, this does not hold for CUB-200, where performance drops below the baseline. Notably, this degradation coincides with a significantly higher reconstruction loss on CUB-200, compared to the other two datasets. FVU > 1 indicates that the model performs worse than simply predicting the mean of the target variable. This leads to a key practical insight: reconstruction loss can be used as a test-time diagnostic to estimate whether steering is likely to be beneficial, an ability notably absent in conventional steering-vector methods, which lack any built-in signal of reliability. When the reconstruction loss of a test-time input is high, a simple fallback to the baseline representation may yield better results.

**Out-of-distribution Steering using Fallback Mechanism.** To evaluate VS2 under distribution shift, we apply the generalized SAE to six unseen datasets: Aircraft (Maji et al., 2013), Food101 (Bossard et al., 2014), Flower102 (Nilsback & Zisserman, 2008), Caltech101(Fei-Fei, 2004), SUN (Xiao et al., 2010), and EuroSAT (Helber et al., 2019). Table 4 summarizes the results. When an in-domain SAE is used, VS2 improves over CLIP on all six datasets. In contrast, the generalized SAE alone can substantially hurt performance on Aircraft and SUN, while still yielding modest gains on Food101, Flower102, and Caltech101. The fallback mechanism exploits the SAE reconstruction loss as a per-sample reliability signal: if the loss exceeds a threshold $\tau$, steering is skipped and the original CLIP embedding is used. With a tight threshold ($\tau = 2.0$), VS2 avoids large negative adaptation on challenging OOD datasets (Aircraft, SUN) while essentially matching the baseline elsewhere. A more detailed sweep over $\tau$ is provided in Appendix O. The fallback mechanism primarily serves as a safety measure and might default to the baseline for OOD datasets; developing more nuanced reliability mechanisms is an interesting research direction for future work.

Table 5: **Reconstruction and classification with a generalized SAE.** We report top-1 accuracy (%) with corresponding reconstruction error FVU in parentheses.

| Method | CIFAR-100 | | CUB-200 | | Tiny-IN | |
|---|---|---|---|---|---|---|
| | B/32 | B/16 | B/32 | B/16 | B/32 | B/16 |
| Baseline | 61.07 (–) | 63.96 (–) | **51.76** (–) | **55.06** (–) | 56.64 (–) | 61.08 (–) |
| VS2 | **64.63** (0.216) | **68.22** (0.208) | 48.81 (1.931) | 49.22 (1.930) | **59.73** (0.448) | **63.83** (0.437) |

## 5 DISCUSSION AND FUTURE WORK

We introduced VS2, a lightweight, label-free test-time method that steers vision models using steering vectors derived from sparse features learned by top-$k$ Sparse Autoencoders (SAEs), without requiring contrastive data as in conventional steering vectors. Our working hypothesis is that SAEs, trained with a self-reconstruction objective, identify salient features; by amplifying these features and forming a sparse steering vector that points toward their direction, VS2 improves classification performance at inference time. This work contributes to the limited literature on *effective downstream use of SAEs*.

The effectiveness of VS2 depends on SAE fidelity: poor reconstructions yield unreliable sparse features and may lead to harmful steering. While our method assumes a foundational SAE that captures the data distribution, test-time inputs may be out-of-distribution. VS2 leverages reconstruction error as a built-in diagnostic which is absent compared to common steering vectors; high error triggers a safe fallback to the zero-shot prediction. One promising research direction includes adapting the SAE per instance at test time via lightweight self-reconstruction to reduce error, improve the salient features, and therefore steering effectiveness.

Diving deeper into why SAEs are effective, we observe an important nuance: features learned for the autoencoding objective are not necessarily relevant for downstream classification. This helps explain why VS2 performs well on general-domain datasets (e.g., CIFAR-100) but is less effective on fine-grained datasets (e.g., CUB-200). A promising direction is to develop SAEs that learn features at multiple levels of granularity, such as Matryoshka SAEs (Bussmann et al., 2025; Zaigrajew et al., 2025), allowing sparse representations at multiple levels.

Most importantly, we identify a central insight: the SAE objective surfaces sparse features for reconstruction, but the features most relevant for a downstream task may differ and are not known a priori. In our current setup, the same SAE activations are produced regardless of the task, while feature relevance is inherently task-dependent. This gap between reconstruction saliency and task saliency motivates future work on task-aligned steering, i.e., constructing steering vectors that align with truly task-salient features. An initial step in this direction appears in the Appendix K with Prototype-Alignment Sparse Steering, inspired by prototype theory (Rosch, 1973), thereby highlighting a promising approach to align SAE-based features with downstream task relevance.

## 6 CONCLUSION

We introduce Visual Sparse Steering VS2 and its retrieval-augmented variant VS2$^{++}$, two test-time methods that steer vision foundation models using sparse features obtained from top-$k$ Sparse Autoencoders without requiring contrastive data. Specifically, VS2 consistently outperforms all baseline methods across all datasets and ViT backbones, surpassing zero-shot CLIP by 3.45% and 4.12% on CIFAR-100, 0.93% and 1.08% on CUB-200, and 1.50% and 1.84% on Tiny-ImageNet with ViT-B/32 and ViT-B/16 backbones, respectively. To control concept importance more precisely, we propose VS2$^{++}$. With oracle positive/negative sets, VS2$^{++}$ surpasses every baseline in nearly every configuration, achieving absolute top-1 gains over CLIP zero-shot of up to 21.44% on CIFAR-100, 7.08% on CUB-200, and 20.47% on Tiny-ImageNet. When those sets are noisy, its accuracy drops but still surpasses the standard Steering Vector in most cases, highlighting the need for better feature selection under weakly supervised retrieval. Interestingly, VS2 and VS2$^{++}$ raise per-class accuracy by up to 25% and 38%, respectively, showing that sparse steering benefits visually or taxonomically proximate classes. Finally, VS2 includes a built-in reliability diagnostic based on SAE reconstruction loss, which can signal when steering may underperform and safely trigger fallback to the baseline.

**Reproducibility Statement.** We reference all details needed to reproduce our results in the paper and appendix. Visual Sparse Steering pseudocode is provided in Appendix G, and training details for top-k Sparse Autoencoders (objectives, hyperparameters, and optimization settings) are in Appendix C. Datasets, zero-shot CLIP backbones, and evaluation metrics are specified in the main text, with hyperparameter sweeps and additional ablations (e.g., sensitivity to $\lambda$, $\gamma$ and layer-wise CLS steering in Appendices B, and E respectively) reported in the main paper and appendix.

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

APPENDIX

# A    DECODING THE SPARSE LATENT SPACE: INSIGHTS FROM SAES

Our proposed methods achieve significant classification performance gains, primarily due to the contribution of Sparse Autoencoders (SAEs). We hypothesize that SAEs identify meaningful sparse features, which in turn guide the steering mechanisms. To validate this assumption, we conduct both quantitative and qualitative evaluations to assess the significance of the learned features.

## A.1    QUANTITATIVE EVALUATION OF FEATURE SIGNIFICANCE

To evaluate the role of sparse features, we manipulate the top-$k$ most active latent features extracted via Sparse Autoencoders. This experiment examines whether these features are critical for classification and how model predictions change under different modifications. We explore the following manipulation settings:

1. **Zeroing-Out** ($\gamma = 0$): We set the top-$k$ most active features to zero before applying the steering vector. This removes their influence while preserving the remaining latent structure.

2. **Negation** ($\gamma = -1$): We invert the sign of the top-$k$ features before applying the steering vector, effectively pushing the representation in the opposite direction. This tests whether these dimensions encode class-discriminative information.

Table 6 reports the zero-shot classification accuracy of $\text{SAE}_{\text{REC}}^{\text{F}+\gamma}$ using ViT-B/32 after applying these transformations to the Sparse Steering vector intervention. **We observe that negating or zeroing the top-$k$ most important features consistently degrades performance across all datasets**. This result confirms that the features learned by the Sparse Autoencoders are essential for classification.

Table 6: **Effect of manipulating top-$k$ sparse codes.** Zero-shot accuracy (%) drops sharply when dominant sparse features are zeroed ($\gamma = 0$) or negated ($\gamma = -1$), confirming their importance.

| Modification | CIFAR-100 | CUB-200 | Tiny-IN |
|---|---|---|---|
| $\text{CLIP}_{\text{ZS}}$ | 61.07 | 38.68 | 56.64 |
| $\text{SAE}_{\text{REC}}^{\text{F}+\gamma}$ | 62.69 | 37.30 | 39.49 |
| *+ Zero-out* ($\gamma = 0$) | 1.71 | 1.14 | 16.11 |
| *+ Negate* ($\gamma = -1$) | 0.06 | 0.00 | 0.82 |

## A.2    QUALITATIVE EVALUATION OF FEATURES SIGNIFICANCE

We qualitatively investigate the features learned in the sparse latent representations of the top-$k$ activations in the Sparse Autoencoder (SAE). Specifically, we assess the learned features by analyzing feature activations for each input and identifying the inputs that exhibit the highest activations for a given feature. Unlike mechanistic interpretability in the language domain, where an LLM can be used to assign semantic labels to a feature by summarizing its highly activated inputs, the vision domain lacks an equivalent automated labeling process.

To avoid reliance on human qualitative evaluation, we leverage annotated datasets where each image is associated with predefined attributes. For the qualitative evaluation of feature significance, we use the CUB dataset, which provides rich concept annotations for each image, enabling a structured assessment of the learned representations. Specifically, we investigate whether we can identify specific latent features with the highest **concept coverage** among their top-k most activated images. Concept coverage refers to how consistently a specific interpretable concept (e.g., an identifiable object category, attribute, or semantic idea) appears across a set of highly activated examples for a given SAE dimension. The intuition is that if a specific concept frequently emerges among the top-activating images for a particular feature, that feature is strongly associated with that concept.

For feature 511 as shown in Figure 2 (left), the activated images exhibit consistent semantic characteristics, including a gray upper part and a white underpart. Notably, this feature predominantly

activates for images from similar but different classes, specifically different types of Gulls, such as Western, California, Herring, and Slaty-Backed Gulls. We observe that the top-activating images for these classes share all concepts in common which is not the case with dimension 3067. For feature 3067, as shown in Figure 2 (right) we observe that the top-activating images share common visual attributes, such as a white-colored throat and black eyes. Through human qualitative evaluation, we find that **features in the sparse latent space capture meaningful visual concepts, grouping semantically similar images together, either from the same class or across different classes, as long as they share underlying conceptual similarities**.

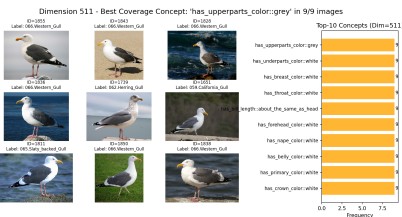 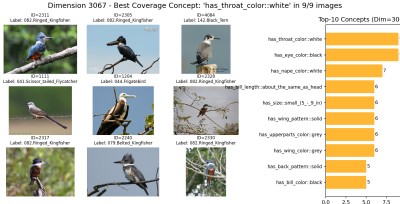

Figure 2: Concept coverage analysis of learned sparse latent features in the Sparse Autoencoder (SAE). Each subfigure illustrates the top-activating images for two different SAE dimensions, highlighting the consistency of shared visual concepts among the highest-activated examples. The analysis demonstrates how sparse features capture meaningful semantic attributes, grouping semantically similar images either within or across classes. **Left**: Top-activating images for feature 511. The images predominantly belong to different classes of Gulls (e.g., Western, California, Herring, and Slaty-Backed Gulls), yet they share consistent visual characteristics such as a gray upper part and a white underpart. This suggests that feature 511 captures a semantically meaningful concept spanning multiple related categories. **Right**: Top-activating images for feature 3067. The images share distinct visual attributes, including a white-colored throat and black eyes. However, unlike feature 511, these images belong to more diverse categories, indicating that this latent dimension captures a broader concept that generalizes across different classes.

**Class-Conditional Feature Activation.** To provide additional evidence that SAE dimensions capture discriminative visual concepts, we report the class-conditional activation frequencies of two example features: 511 and 3067. Specifically, Table **??** shows the percentage of samples in the top-10 activating classes where each feature is present.

Table 7: Top-10 most activated classes for SAE Feature 511 (left) and Feature 3067 (right). Feature 511 strongly aligns with gull-like classes, while Feature 3067 captures a cross-class head/throat attribute.

| Class (511) | Activation | Class (3067) | Activation |
|---|---|---|---|
| 066.Western_Gull | 96.7% (29/30) | 082.Ringed_Kingfisher | 60.0% (18/30) |
| 060.Glaucous_winged_Gull | 96.6% (28/29) | 083.White_breasted_Kingfisher | 40.0% (12/30) |
| 061.Heermann_Gull | 93.3% (28/30) | 079.Belted_Kingfisher | 30.0% (9/30) |
| 059.California_Gull | 90.0% (27/30) | 041.Scissor_tailed_Flycatcher | 23.3% (7/30) |
| 062.Herring_Gull | 83.3% (25/30) | 080.Green_Kingfisher | 16.7% (5/30) |
| 063.Ivory_Gull | 76.7% (23/30) | 002.Laysan_Albatross | 13.3% (4/30) |
| 087.Mallard | 76.7% (23/30) | 025.Pelagic_Cormorant | 13.3% (4/30) |
| 147.Least_Tern | 76.7% (23/30) | 025.Pelagic_Cormorant | 13.3% (4/30) |
| 064.Ring_billed_Gull | 70.0% (21/30) | 044.Frigatebird | 10.0% (3/30) |
| 084.Red_legged_Kittiwake | 73.9% (17/23) | 049.Boat_tailed_Grackle | 10.0% (3/30) |

**Class-Conditional Feature Activation.** To provide additional evidence that SAE dimensions capture discriminative visual concepts, we report the class-conditional activation frequencies of two example features: 511 and 3067. Specifically, Table 7 shows the percentage of samples in the top-10 activating classes where each feature is present.

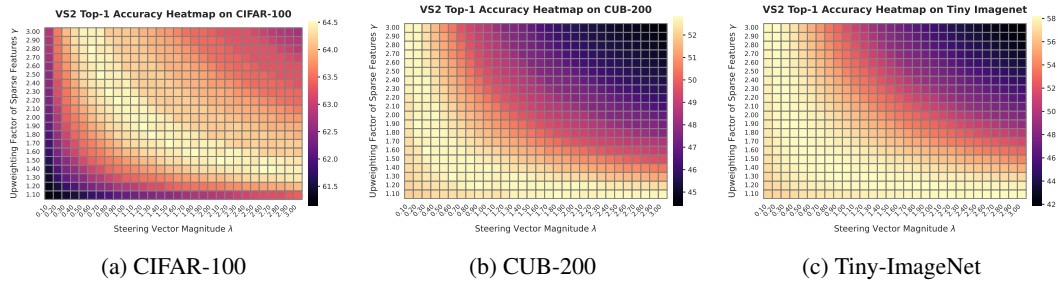

(a) CIFAR-100        (b) CUB-200        (c) Tiny-ImageNet

Figure 3: **Sensitivity of VS2 to sparse amplification** $\gamma$ **and steering magnitude** $\lambda$. All three datasets show a range of near-optimal combinations (warm colours), typically when $\lambda \cdot \gamma \in [2, 3]$. Accuracy degrades if either parameter becomes too large.

What we observe is that feature 511 fires very often in gull-like classes. It also fires Larus-type seabirds, i.e., terns, kittiwakes. We observe that it also fires moderately on similar seabirds (Horned Puffin: 53.3%, White Pelican: 40%, Brown Pelican: 16.7%, Red-breasted Merganser: 10%). Additionally, we observe that Feature 511 activates near 0% on unrelated classes such as buntings, warblers, and hummingbirds. In contrast, Feature 3067 captures a cross-category visual attribute rather than a taxonomic grouping. Specifically, it consistently activates on species that share a distinctive head-and-throat pattern (e.g., white throat with darker eye region), even when those species are taxonomically unrelated. This indicates that the SAE possibly learns attribute-level discriminative concepts, complementing the class-specific features exemplified by Feature 511.

**Discussion.** We note that our goal is not to claim human interpretability of all SAE dimensions. As shown in recent mechanistic interpretability work (**?**), human-aligned features are often split across multiple SAE components (feature absorption), a known behavior in sparse dictionary learning. Nonetheless, our contribution lies in demonstrating that SAEs can go beyond interpretability, serving as effective mechanisms for performance improvement via sparse steering. While some features are interpretable, our main result is that sparse features enable controllable improvements in downstream classification tasks.

## B    SENSITIVITY TO SPARSE AMPLIFICATION ($\gamma$) AND STEERING MAGNITUDE ($\lambda$)

We analyze how the two key hyperparameters in VS2 (i) the sparse-feature amplification $\gamma$, and (ii) the steering vector scale $\lambda$ affect downstream accuracy. In the absence of contrastive supervision, these parameters govern how strongly we amplify sparse activations and how far the embedding is shifted in feature space. We sweep both values across a grid on three datasets: CIFAR-100, CUB-200, and Tiny-ImageNet using ViT-B/32 backbone.

Figure 3 shows that all tested combinations of $(\lambda, \gamma)$ outperform the zero-shot baseline, though to varying degrees. Each dataset exhibits a diagonal band of near-optimal settings where $\lambda \cdot \gamma \in [2, 3]$ tends to yield peak accuracy. For example, CIFAR-100 peaks at $\lambda^* = 2.1$ and $\gamma^* = 1.5$. Beyond this band, increasing $\lambda$ or $\gamma$ causes performance to degrade likely due to over-amplification of sparse features and/or embedding distortion. The consistent contour patterns across datasets suggest that VS2 is robust to moderate variation in its hyperparameters and that good settings generalize well across domains.

## C    TRAINING TOP-$k$ SPARSE AUTOENCODERS

We follow CLIP (Radford et al., 2021b) with a ViT-B/32 and ViT-B/16 vision backbones, intercepting the output of each encoder layer for the $CLS$ token. Specifically, we train top-$k$ SAEs on the $CLS$ embeddings for each chosen layer. We use $k = 64$ and $k = 256$ as the maximum active features within the latent space for ViT-B/32 and ViT-B/16 respectively, and we set a "dead feature" threshold of 100 i.e., any feature seldom activated is pruned. We also use an expansion factor of 4 relative to

Table 8: **Fraction of variance unexplained (FVU; lower is better).** Each SAE has 4× expansion; sparsity $k = 64$ for ViT-B/32 and $k = 256$ for ViT-B/16.

| Dataset | ViT-B/32 (k=64) | ViT-B/16 (k=256) |
|---------|-----------------|-------------------|
| CIFAR-100 | 0.2812 | 0.1166 |
| CUB-200 | 0.2487 | 0.1653 |
| Tiny-ImageNet | 0.5060 | 0.3018 |

the input embedding dimension, resulting in 3,072 latent units. Training largely follows Gao et al. (2024a) and uses EleutherAI (2024), with a linear learning-rate schedule and warmup from $5 \times 10^{-4}$.

We monitor reconstruction quality with the **fraction of variance unexplained** (**FVU**; lower is better), defined as

$$\text{FVU} \;=\; \frac{\|\mathbf{X} - \hat{\mathbf{X}}\|_F^2}{\|\mathbf{X} - \bar{\mathbf{X}}\|_F^2}, \tag{5}$$

where $\mathbf{X} \in \mathbb{R}^{B \times d}$ is a batch of CLS embeddings, $\hat{\mathbf{X}}$ denotes their SAE reconstructions, and $\bar{\mathbf{X}}$ is the batch mean (so the denominator equals the total variance). FVU is the complement of the coefficient of determination $(1 - R^2)$; an FVU of 0 indicates perfect reconstruction, while an FVU of 1 corresponds to predicting only the mean. In Table 8, we report FVU results for ViT-B/32 with $k = 64$ and ViT-B/16 with $k = 256$ across all three datasets.

## D  SENSITIVITY IN RETRIEVAL-AUGMENTED GENERATION (RAG)

When an external image corpus is available, VS2 can be extended using a Retrieval-Augmented Generation (RAG) pipeline. We use DINOv2 Oquab et al. (2023) to retrieve top-$k$ images most similar to the input query and compute an enhanced embedding by combining the query with the retrieved set:

$$\mathbf{E} = \alpha \mathbf{q} + (1 - \alpha) \sum_{j=1}^{k} w_j \mathbf{r}_j,$$

where $\mathbf{q}$ is the query embedding, $\mathbf{r}_j$ the $j$-th retrieved embedding, and $w_j$ the normalized similarity weight:

$$w_j = \frac{s_j}{\sum_{i=1}^{k} s_i}, \quad \text{where } s_j = \text{sim}(\mathbf{q}, \mathbf{r}_j).$$

The parameter $\alpha \in [0, 1]$ controls the trade-off between using the original query and the retrieved set. We sweep over values of $\alpha$ and $k$ to assess their impact on zero-shot classification performance. Figure 4 shows results for CIFAR-100, and Tiny-ImageNet. CIFAR-100 and Tiny-ImageNet display a similar trend: larger $\alpha$ (i.e., more reliance on the query) typically degrades performance. On Tiny-ImageNet, setting $\alpha = 0$, completely ignoring the input query and relying purely on retrieved features, yields the best result. Across all datasets, large $k$ eventually hurts performance, confirming that RAG benefits from focused rather than broad context. The trade-off parameter $\alpha$ and the retrieval depth $k$ play dataset-dependent roles in RAG-enhanced pipelines. For fine-grained domains, fewer, high-confidence neighbors and a low $\alpha$ might work best; for noisy domains, more reliance on the retrieval images might be more beneficial.

## E  STEERING ACROSS LAYERS OF THE CLS TOKEN

Our default method reconstructs and applies steering to the CLS token embedding at the final layer of the vision encoder. To study depth effects, we evaluate steering the CLS token across multiple final layers of the transformer. We use ViT-B/16 on CIFAR-100 with expansion factor 4 and 128 sparse activations. Table 9 reports top-1 accuracy when steering the last 1, 2, or 3 layers. Steering only the final layer yields the best performance. Adding earlier layers causes a progressive drop in accuracy, falling below the zero-shot baseline when steering the last three layers.

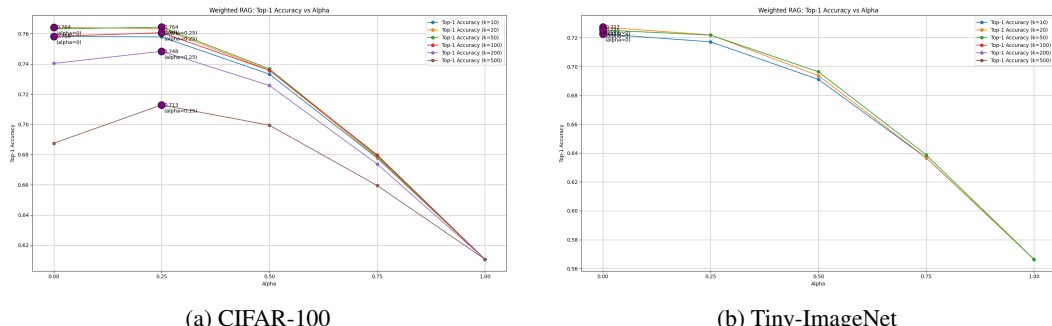

(a) CIFAR-100             (b) Tiny-ImageNet

Figure 4: **RAG sensitivity to $\alpha$ and top-$k$.** Accuracy varies with the weight $\alpha$ on the original query and the number $k$ of retrieved images. Larger $k$ often introduces noise; smaller $\alpha$ performs better on cluttered datasets.

Table 9: **Effect of steering depth on CLS.** Top-1 accuracy (%) on CIFAR-100 (ViT-B/16) as the number of steered final layers increases.

| Steered Layers | Accuracy (%) |
| --- | --- |
| 12 | 68.08 |
| 11 + 12 | 65.72 |
| 10 + 11 + 12 | 59.36 |

Applying steering at multiple layers likely introduces compounding perturbations that propagate forward, making later representations harder to align with the classifier. CLS steering is most effective at the final layer. Future work could explore coordinated multi-layer steering to avoid error accumulation.

# F  ABLATION STUDY OF EXPANSION FACTOR AND TOP-$k$

In Table 10, we present the downstream task accuracy of CLIP ViT-B/32 using various values of expansion factor and $k$. Across a $4 \times$ range in width and an $8 \times$ range in sparsity, top-1 accuracy fluctuates by less than one percentage point *evidence that VS2's performance is largely insensitive to the precise SAE capacity–sparsity trade-off.* Additionally, in Table 16, we present the average cosine similarities of different steering vectors coming from various configurations of SAEs in terms of expansion factor and top-$k$.

# G  VISUAL SPARSE STEERING PSEUDOCODE

For reproducibility purposes, in Algorithm 1, we provide the pseudocode for the baselines and VS2 used in the analysis of 4.1.

Table 10: **VS2 accuracy as a function of SAE width (expansion factor) and sparsity (top-$k$).** Numbers are top-1 / top-5 (%). The best result is boldfaced; every other configuration is within of the optimum, highlighting the method's robustness to architectural choices.

| SAE configuration | Top-1 ↑ | Top-5 ↑ |
|---|---|---|
| $4\times, k{=}128$ | **64.61** | **87.95** |
| $8\times, k{=}128$ | 64.56 | 87.76 |
| $4\times, k{=}64$ | 64.54 | 87.79 |
| $16\times, k{=}64$ | 64.54 | 87.78 |
| $8\times, k{=}64$ | 64.52 | 87.96 |
| $10\times, k{=}128$ | 64.43 | 87.81 |
| $16\times, k{=}512$ | 64.42 | 87.71 |
| $8\times, k{=}512$ | 64.40 | 87.91 |
| $4\times, k{=}256$ | 64.34 | 87.62 |
| $8\times, k{=}256$ | 64.29 | 87.80 |
| $16\times, k{=}256$ | 64.28 | 87.75 |
| $4\times, k{=}512$ | 64.12 | 87.87 |
| $16\times, k{=}128$ | 64.10 | 87.79 |

---

**Algorithm 1** SAE_STEERING – SAE-based CLS token modification

---

1: **function** SAE_STEERING($\mathbf{h}$, SAE, $k$, $\gamma$, $\lambda_{\Delta z}$, mode) ▷ $\mathbf{h}$: CLS token; SAE: sparse autoencoder;
   $k$: sparsity level
2:     $\mathbf{a}, \mathbf{idx} \leftarrow$ SAE.select_topk(SAE.pre_acts($\mathbf{h}$), $k$)
3:     $\mathbf{z}_{\text{base}} \leftarrow$ SAE.decode($\mathbf{a}, \mathbf{idx}$)
4:     $\mathbf{z}_{\text{boost}} \leftarrow$ SAE.decode($\gamma \cdot \mathbf{a}, \mathbf{idx}$)
5:     **if** mode = "reconstruction" **then**
6:         **return** $\mathbf{z}_{\text{base}}$                                          ▷ Variant: $\text{CLIP}_{\text{REC}}^{\text{F}}$
7:     **else if** mode = "amplified" **then**
8:         **return** $\mathbf{z}_{\text{boost}}$                                          ▷ Variant: $\text{CLIP}_{\text{REC}}^{\text{F}+\gamma}$
9:     **else if** mode = "steering" **then**
10:        **return** $\mathbf{h} + \lambda_{\Delta z} \cdot (\mathbf{z}_{\text{boost}} - \mathbf{z}_{\text{base}})$                      ▷ VS2
11:    **end if**
12: **end function**

---

## H COMPARISON WITH OTHER BASELINE METHODS

We compare VS2 against SpLiCE (Bhalla et al., 2024), using the official implementation provided by the authors. For all three datasets, we report performance using SpLiCE with an external vocabulary of 10,000 LAION-based concepts and an $\ell_1$ regularization weight of 0.25, following their best reported configuration. Despite relying on no external vocabulary, VS2 consistently outperforms SpLiCE across all benchmarks highlighting the strength of sparse concept steering even in the absence of external lexical resources.

Table 11: Zero-shot top-1 accuracy (%) on **CIFAR-100**.

| Method | ViT-B/32 | ViT-B/16 |
|---|---|---|
| $\text{CLIP}_{\text{ZS}}$ | 61.07 ( 0 ) | 63.96 ( 0 ) |
| $\text{SAE}_{\text{REC}}^{\text{A}}$ | 58.01 ( −3.06 ) | 64.05 ( +0.09 ) |
| $\text{SAE}_{\text{REC}}^{\text{F}}$ | 58.22 ( −2.85 ) | 63.42 ( −0.54 ) |
| $\text{SAE}_{\text{REC}}^{\text{F}+\gamma}$ | 62.69 ( +1.62 ) | 66.81 ( +2.85 ) |
| SpLiCE (Bhalla et al., 2024) | 55.57 ( −5.50 ) | 58.29 ( −5.67 ) |
| **VS2 (ours)** | **64.52** ( **+3.45** ) | **68.08** ( **+4.12** ) |

## I    EFFECT OF TOP-$N$ RETRIEVED NEIGHBORS

To assess the influence of retrieval size on performance, we conduct an ablation over the number of retrieved neighbors $N$ used for latent aggregation. We use a fixed ViT-B/16 (Patch-16) backbone and vary $N \in \{10, 25, 50, 100\}$ across three datasets: CIFAR-100, CUB-200, and Tiny-ImageNet. As shown in Figure 5, both top-1 and top-5 classification accuracy steadily increase with $N$ on CIFAR-100 and Tiny-ImageNet, with diminishing returns beyond $N = 50$. In contrast, performance on CUB-200 remains largely flat or slightly degrades, suggesting that retrieving more neighbors in fine-grained datasets can introduce noise due to overly similar but semantically irrelevant examples. These results indicate that the utility of neighbor-based aggregation is dataset-dependent: general object recognition tasks may benefit from larger $N$, while fine-grained classification may require more careful control of retrieval scope.

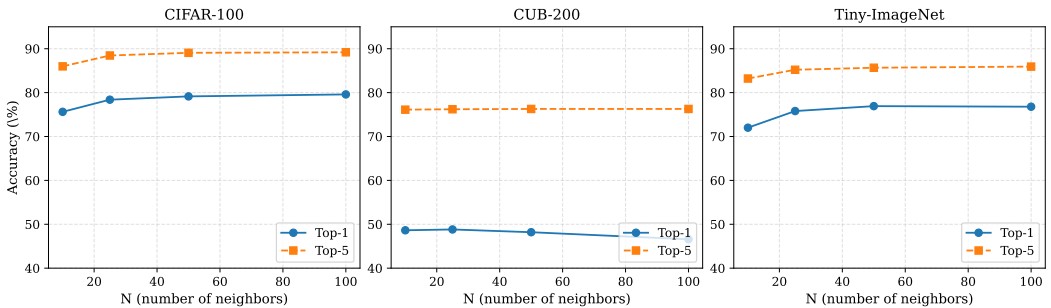

Figure 5: Top-1 and Top-5 accuracy as a function of number of retrieved neighbors $N$, using ViT-B/16 (Patch-16) across datasets. Larger $N$ improves general classification but may degrade performance in fine-grained settings.

## J    PER-CLASS PERFORMANCE ANALYSIS ON CIFAR-100

In Table 15, we present results for the classes most affected by steering CLIP ViT-B/32 with VS2 and VS2$^{++}$. We highlight both positive and negative shifts in accuracy to identify the most impacted categories. Unlike the average performance reported in the main experiments, this analysis shows that the top-10 class-level gains are generally larger in magnitude than the corresponding losses. Although the accuracy drops for misclassified categories are smaller in magnitude, they are non-negligible. As future work, it would be valuable to explore when steering should be applied or withheld to further maximize overall performance gains.

## K    PROTOTYPE-AWARE SPARSE STEERING VECTORS

Our core methods, VS2 and its retrieval-augmented variant VS2$^{++}$, enhance zero-shot classification by steering CLIP embeddings along directions identified by a top-$k$ Sparse Autoencoder (SAE). These directions correspond to latent *features* that, ideally, align with class-discriminative concepts. Steering in these directions upweights what the model has learned to be important during reconstruction. This raises a central hypothesis: **the reconstruction task itself is to some extent sufficient to uncover features that are also relevant for classification**. In other words, there is a meaningful overlap between features that are important for reconstructing the CLS token and those that are predictive for the downstream task. In this section, drawing inspiration from prototype theory (Rosch, 1973), we investigate whether incorporating prototype information during SAE training can better align the features important for reconstruction with those that are critical for downstream classification.

The limited improvements observed on fine-grained datasets like CUB-200 and Tiny-ImageNet suggest that the challenge lies not just in identifying sparse features, but in uncovering the *correct* ones. This shifts the central question from "What are the most important sparse features to select?" to a deeper inquiry: "Can the reconstruction objective alone reliably capture features that are most useful for classification and if not, how can task-relevant information be effectively incorporated?"

Table 12: Benchmarking Visual Sparse Steering: Zero-Shot Accuracy (%) with and Without External Data on CIFAR-100, CUB-200, and Tiny Imagenet using ViT-B/32 and ViT-B/16.

| Method | CIFAR-100 | | CUB-200 | | Tiny-IN | |
|---|---|---|---|---|---|---|
| | ViT-B/32 | ViT-B/16 | ViT-B/32 | ViT-B/16 | ViT-B/32 | ViT-B/16 |
| **Zero-shot (no retrieval)** | | | | | | |
| $CLIP_{ZS}$ | 61.07 ( 0 ) | 63.96 ( 0 ) | 51.76 ( 0 ) | 55.06 ( 0 ) | 56.64 ( 0 ) | 61.08 ( 0 ) |
| $SAE_{REC}^{A}$ | 58.01 ( −3.06 ) | 64.05 ( +0.09 ) | 47.45 ( -4.31 ) | 51.81 ( -3.25 ) | 30.56 ( −26.08 ) | 52.96 ( -8.12 ) |
| $SAE_{REC}^{F}$ | 58.22 ( −2.85 ) | 63.42 ( −0.54 ) | 48.08 ( -3.68 ) | 51.43 ( -3.63 ) | 36.33 ( −20.31 ) | 54.84 ( -6.24 ) |
| $SAE_{REC}^{F+\gamma}$ | 62.69 ( +1.62 ) | 66.81 ( +2.85 ) | 49.41 ( -2.35 ) | 53.28 ( -1.78 ) | 39.49 ( −17.15 ) | 58.81 ( -2.27 ) |
| **VS2 (ours)** | **64.52** ( +3.45 ) | **68.08** ( +4.12 ) | **52.69** ( +0.93 ) | **56.14** ( +1.08 ) | **58.14** ( +1.50 ) | **62.92** ( +1.84 ) |
| **VS2 + PASS (ours)** | **70.64** ( +9.57 ) | **68.23** ( +4.27 ) | **52.97** ( +1.21 ) | **56.63** ( +1.57 ) | **57.94** ( +1.30 ) | **62.98** ( +1.90 ) |

**Oracle steering with known prototypes.** Following prototype theory, we collect for every class the ten images that ViT-B/32 CLIP classifies with the highest confidence; these serve as *oracle prototypes*. Given the true class label $y$, we build a prototype steering vector by averaging their latent sparse features. **VS2 using oracle prototypes can lift CLIP to 97.5% on CIFAR-100, 91.04% on CUB-200, and 90.1% on Tiny-ImageNet**. This confirms the existence of discriminative sparse directions. In Appendix L we further examine the discriminative ability (measured by orthogonality) of these steering vectors. Generally, these prototype vectors have a low cosine similarity, yet a non-negligible tail reveals strongly overlapping directions between visually or semantically close categories.

**Prototype-aligned SAE (PASS).** Above, we constructed steering vectors using the pretrained SAE features of oracle prototypes. Now, we consider whether prototypes can be used to learn new, more informative SAE features. We assume during SAE training that we have access to class labels. Then, to the SAE loss we add a regularization term which encourages SAE features to be close to their class mean. That is, for a training sample $i$ with latent sparse feature $\mathbf{z}_i$ and class mean $\bar{\mathbf{z}}_{\text{class}(i)}$ we minimize

$$\mathcal{L} = \mathcal{L}_{\text{recon}} + w_{\text{aux}} \left\| \mathbf{z}_i - \bar{\mathbf{z}}_{\text{class}(i)} \right\|_2^2, \tag{6}$$

where $w_{\text{aux}}$ controls the strength of the prototype-alignment term relative to the reconstruction loss and is set to 0.8. We refer to the resulting steering method as **PASS** (Prototype-Aligned Sparse Steering). Although PASS uses class labels during SAE training, it remains *fully test-time unsupervised*. Empirically, in Table 12, we observe that PASS outperforms VS2 across all datasets, with particularly substantial gains on CIFAR-100. However, this improvement comes at the cost of requiring labels for each training sample during SAE training. Gains are modest on CUB-200 and Tiny-ImageNet, and we thus hypothesize that classes which share many features require richer or multi-prototype guidance which is an intriguing avenue for future work.

Table 13: **Trade-off between reconstruction fidelity and prototype alignment.** Increasing $w_{\text{aux}}$ improves classification accuracy but degrades FVU reconstruction.

| $w_{\text{aux}}$ | FVU $\downarrow$ | Accuracy (%) $\uparrow$ |
|---|---|---|
| 0.1 | 0.3437 | 68.80 |
| 0.5 | 0.4887 | 70.40 |
| 1.0 | 0.5393 | 70.61 |
| 2.0 | 0.5702 | 70.76 |

**Reconstruction vs. Alignment Trade-off** Sparse Autoencoders (SAEs) trained for reconstruction can also be optimized to align their latent features with class-level prototypes. However, this introduces a trade-off between two competing objectives: fidelity of reconstruction and discriminative alignment.

To investigate this trade-off, we introduce a weighting coefficient $w_{\text{aux}}$ that controls the strength of prototype alignment relative to the reconstruction loss. As $w_{\text{aux}}$ increases, alignment is encouraged more strongly. Table 13 reports the resulting changes in reconstruction loss (measured by FVU) and

top-1 classification accuracy on CIFAR-100 using ViT-B/16, with 128 sparse latents and expansion factor 4.

We observe that increasing $w_{\text{aux}}$ consistently improves classification performance, up to a point, even though it introduces more reconstruction error. This aligns with our hypothesis: while exact input reconstruction encourages general feature coverage, alignment with class prototypes promotes discriminative feature extraction. These results highlight the flexibility of SAE-based steering to balance interpretability and performance depending on downstream objectives.

## L    ARE EXEMPLAR-DERIVED DIRECTIONS REALLY DISTINCT?

A desirable property of class–specific steering vectors is *orthogonality*: pushing an embedding toward class *A* should not simultaneously raise its score for class *B*. Using the oracle prototypes as described in Appendix K, we compute for every CIFAR-100 class a prototype steering vector and measure the pair-wise cosine similarity at layer 11 of the ViT-B/32 encoder. Most pairs have low similarity (mean=0.23), yet a non-negligible tail reveals strongly overlapping directions. Table 14 lists the ten highest-overlap pairs.

Table 14: **Top-10 most overlapping prototype steering directions on CIFAR-100**. High cosine similarity indicates that the two classes share visual attributes that the SAE encodes along nearly the same latent axis.

| Rank | Class 1 | Class 2 | Cosine ↑ |
|------|---------|---------|----------|
| 1 | beetle | cockroach | 0.91 |
| 2 | mouse | shrew | 0.89 |
| 3 | dolphin | shark | 0.84 |
| 4 | otter | seal | 0.84 |
| 5 | dolphin | whale | 0.84 |
| 6 | possum | raccoon | 0.84 |
| 7 | snake | worm | 0.83 |
| 8 | oak tree | willow tree | 0.83 |
| 9 | ray | shark | 0.81 |
| 10 | bowl | cup | 0.80 |

These high-overlap pairs are *semantically plausible* confusions (e.g. beetle vs. cockroach or dolphin vs. whale), confirming that exemplar steering directions tend to align for visually or taxonomically proximate classes. In downstream applications, a simple orthogonalization step may help reduce feature overlap between sparse directions. Investigating principled ways to encourage orthogonality during SAE training is a promising direction for future work.

## M    USE OF LLMS

We used Large Language Models (LLMs) to refine the paper text for grammar, syntax, and spelling errors.

## N    COMPUTATIONAL OVERHEAD: ADDITIONAL DETAILS

**Training-time FLOPs.**    To quantify the cost of training the SAE versus alternative adaptation methods, Table 17 reports per-sample training FLOPs (forward + backward) and the number of trainable parameters for the CLIP ViT-B/32 vision encoder.

Even when activation extraction is included, SAE training remains more than $3\times$ cheaper than LoRA or full fine-tuning. With cached activations, as in our VS2 setup, the cost drops to just 0.014 GFLOPs per sample.

**TPT FLOP accounting.**    For completeness, we outline how the $16.6\times$ test-time overhead factor for TPT (Shu et al., 2022) is obtained. A single CLIP ViT-B/32 forward pass on a $224 \times 224$ image with

Table 15: **Top-10 class gains and losses on CIFAR-100.** Green = absolute gain; Red = absolute loss relative to ZS baseline.

(a) **VS2 – Top-10 Gains**

| Class | ZS | VS2 ↑(Δ) |
|---|---|---|
| tractor | 0.55 | 0.80 ↑(+0.25) |
| forest | 0.35 | 0.58 ↑(+0.23) |
| man | 0.53 | 0.74 ↑(+0.21) |
| bus | 0.51 | 0.68 ↑(+0.17) |
| snake | 0.61 | 0.76 ↑(+0.15) |
| woman | 0.57 | 0.71 ↑(+0.14) |
| trout | 0.18 | 0.32 ↑(+0.14) |
| cockroach | 0.49 | 0.62 ↑(+0.13) |
| bridge | 0.72 | 0.83 ↑(+0.11) |
| rose | 0.58 | 0.68 ↑(+0.10) |

(b) **VS2 – Top-10 Losses**

| Class | ZS | VS2 (Δ) |
|---|---|---|
| aquarium_fish | 0.82 | 0.61 (-0.21) |
| beetle | 0.64 | 0.49 (-0.15) |
| sweet_pepper | 0.70 | 0.58 (-0.12) |
| tulip | 0.78 | 0.71 (-0.07) |
| maple_tree | 0.57 | 0.50 (-0.07) |
| flatfish | 0.55 | 0.49 (-0.06) |
| willow_tree | 0.46 | 0.41 (-0.05) |
| lamp | 0.75 | 0.70 (-0.05) |
| lawn_mower | 1.00 | 0.96 (-0.04) |
| kangaroo | 0.72 | 0.68 (-0.04) |

(c) **VS2++ – Top-10 Gains**

| Class | ZS | VS2++ ↑(Δ) |
|---|---|---|
| spider | 0.48 | 0.91 ↑(+0.43) |
| caterpillar | 0.27 | 0.68 ↑(+0.41) |
| possum | 0.24 | 0.65 ↑(+0.41) |
| tractor | 0.55 | 0.96 ↑(+0.41) |
| tiger | 0.45 | 0.84 ↑(+0.39) |
| raccoon | 0.44 | 0.83 ↑(+0.39) |
| lizard | 0.36 | 0.74 ↑(+0.38) |
| wolf | 0.54 | 0.90 ↑(+0.36) |
| shrew | 0.44 | 0.79 ↑(+0.35) |
| bear | 0.51 | 0.85 ↑(+0.34) |

(d) **VS2++ – Top-10 Losses**

| Class | ZS | VS2++ (Δ) |
|---|---|---|
| girl | 0.72 | 0.65 (-0.07) |
| maple_tree | 0.57 | 0.51 (-0.06) |
| porcupine | 0.18 | 0.13 (-0.05) |
| ray | 0.06 | 0.02 (-0.04) |
| mouse | 0.18 | 0.15 (-0.03) |
| lawn_mower | 1 | 0.98 (-0.02) |

100 text prompts costs 8.7 GFLOPs for the vision encoder and 581.8 GFLOPs for the text encoder, for a total of approximately 0.59 TFLOPs. In our implementation, TPT performs 63 augmented views per image and runs 4 optimization steps; each step requires a full CLIP forward pass over all views and a backward pass through the text encoder. This yields about 9.217 TFLOPs of adaptation compute per test image, followed by one final forward-only evaluation of 0.591 TFLOPs, for a total of roughly 9.807 TFLOPs. Dividing by the 0.59 TFLOPs of plain CLIP gives the reported ≈ 16.6× test-time compute overhead.

## O   FALLBACK THRESHOLD SWEEP

Table 18 provides the full sweep over reconstruction-loss thresholds used by the fallback mechanism when steering with the generalized SAE (trained on CIFAR-100, CUB-200, and Tiny-ImageNet) on six unseen datasets. For each threshold $\tau$, VS2 is applied only to samples whose reconstruction loss is below $\tau$; otherwise, we revert to the CLIP zero-shot prediction.

Across all six datasets, small thresholds (e.g., $\tau \in [1.5, 3.0]$) are sufficient to avoid significant performance drops relative to CLIP, effectively preventing harmful steering when the generalized

SAE reconstructs poorly. Larger thresholds eventually reintroduce the failures of the generalized SAE, particularly on Aircraft and SUN, underscoring the importance of a conservative fallback in OOD regimes.

Table 16: **Cosine similarity (↑) between steering vectors learned by different SAE capacities.** Rows/columns are ordered by expansion factor and sparsity (e × expansion, k active).

| SAE cfg. | 16×/64 | 16×/512 | 16×/128 | 16×/256 | 8×/128 | 8×/512 | 8×/256 | 4×/256 | 4×/512 | 4×/128 | 10×/128 | 4×/64 | 8×/64 |
|---|---|---|---|---|---|---|---|---|---|---|---|---|---|
| 16×/64 | **1.000** | 0.497 | 0.577 | 0.705 | 0.726 | 0.635 | 0.680 | 0.640 | 0.591 | 0.681 | 0.536 | 0.701 | 0.716 |
| 16×/512 | 0.497 | **1.000** | 0.161 | 0.399 | 0.358 | 0.817 | 0.634 | 0.618 | 0.611 | 0.584 | 0.136 | 0.316 | 0.282 |
| 16×/128 | 0.577 | 0.161 | **1.000** | 0.900 | 0.865 | 0.273 | 0.298 | 0.277 | 0.265 | 0.303 | 0.952 | 0.887 | 0.678 |
| 16×/256 | 0.705 | 0.399 | 0.900 | **1.000** | 0.956 | 0.527 | 0.572 | 0.495 | 0.410 | 0.542 | 0.920 | 0.936 | 0.744 |
| 8×/128 | 0.726 | 0.358 | 0.865 | 0.956 | **1.000** | 0.543 | 0.650 | 0.575 | 0.483 | 0.634 | 0.896 | 0.959 | 0.829 |
| 8×/512 | 0.635 | 0.817 | 0.273 | 0.527 | 0.543 | **1.000** | 0.861 | 0.811 | 0.771 | 0.796 | 0.243 | 0.483 | 0.506 |
| 8×/256 | 0.680 | 0.634 | 0.298 | 0.572 | 0.650 | 0.861 | **1.000** | 0.889 | 0.771 | 0.912 | 0.289 | 0.557 | 0.624 |
| 4×/256 | 0.640 | 0.618 | 0.277 | 0.495 | 0.575 | 0.811 | 0.889 | **1.000** | 0.873 | 0.941 | 0.240 | 0.532 | 0.600 |
| 4×/512 | 0.591 | 0.611 | 0.265 | 0.410 | 0.483 | 0.771 | 0.771 | 0.873 | **1.000** | 0.821 | 0.191 | 0.461 | 0.549 |
| 4×/128 | 0.681 | 0.584 | 0.303 | 0.542 | 0.634 | 0.796 | 0.912 | 0.941 | 0.821 | **1.000** | 0.284 | 0.594 | 0.655 |
| 10×/128 | 0.536 | 0.136 | 0.952 | 0.920 | 0.896 | 0.243 | 0.289 | 0.240 | 0.191 | 0.284 | **1.000** | 0.910 | 0.678 |
| 4×/64 | 0.701 | 0.316 | 0.887 | 0.936 | 0.959 | 0.483 | 0.557 | 0.532 | 0.461 | 0.594 | 0.910 | **1.000** | 0.814 |
| 8×/64 | 0.716 | 0.282 | 0.678 | 0.744 | 0.829 | 0.506 | 0.624 | 0.600 | 0.549 | 0.655 | 0.678 | 0.814 | **1.000** |

Table 17: Training-time compute and parameter cost per sample for various adaptation strategies on the CLIP ViT-B/32 vision encoder.

| Method | Train FLOPs / sample (fwd+bwd) | GFLOPs | Trainable Params |
|---|---|---|---|
| SAE (cached activations) | 14,164,992 | 0.014 | 4,722,432 |
| SAE (non-cached; incl. extraction) | 8,743,645,440 | 8.744 | 4,722,432 |
| LoRA (rank = 16) | 26,365,388,544 | 26.365 | 589,824 |
| Full fine-tuning (vision encoder) | 26,188,441,344 | 26.188 | 87,456,000 |

Table 18: Out-of-distribution performance of VS2 using a generalized SAE trained on CIFAR-100, CUB-200, and Tiny-ImageNet. We report top-1 accuracy (%) with gains over the CLIP baseline in parentheses. Tight thresholds (e.g., $\tau \leq 2.0$) avoid degradation relative to the baseline.

| Method / Threshold | Aircraft | Food101 | Flower102 | Caltech101 | SUN | EuroSAT |
|---|---|---|---|---|---|---|
| Baseline | 24.84 | 80.71 | 63.72 | 78.53 | 51.79 | 31.61 |
| VS2 (In-domain SAE) | 27.69 | 81.20 | 64.81 | 80.95 | 54.80 | 34.65 |
|  | (+2.85) | (+0.49) | (+1.09) | (+2.42) | (+3.01) | (+3.04) |
| VS2 (Generalized SAE) | 18.57 | 81.06 | 64.11 | 79.72 | 45.78 | 30.76 |
|  | (-6.27) | (+0.35) | (+0.39) | (+1.19) | (-6.01) | (-0.85) |
| VS2 + Fallback (Generalized SAE) |  |  |  |  |  |  |
| Threshold = 1.5 | 24.87 | 80.72 | 63.70 | 78.53 | 51.79 | 31.61 |
|  | (+0.03) | (+0.01) | (-0.02) | (+0.00) | (+0.00) | (+0.00) |
| Threshold = 2.0 | 24.87 | 80.72 | 63.70 | 78.53 | 51.79 | 31.61 |
|  | (+0.03) | (+0.01) | (-0.02) | (+0.00) | (+0.00) | (+0.00) |
| Threshold = 3.0 | 24.87 | 80.72 | 63.70 | 78.53 | 51.79 | 31.61 |
|  | (+0.03) | (+0.01) | (-0.02) | (+0.00) | (+0.00) | (+0.00) |
| Threshold = 4.0 | 24.87 | 80.71 | 63.88 | 78.11 | 51.79 | 30.41 |
|  | (+0.03) | (+0.00) | (+0.16) | (-0.42) | (+0.00) | (-1.20) |
| Threshold = 5.0 | 24.12 | 80.81 | 64.16 | 79.39 | 49.11 | 30.76 |
|  | (-0.72) | (+0.10) | (+0.44) | (+0.86) | (-2.68) | (-0.85) |
| Threshold = 6.5 | 18.69 | 81.05 | 64.11 | 79.72 | 45.86 | 30.76 |
|  | (-6.15) | (+0.34) | (+0.39) | (+1.19) | (-5.93) | (-0.85) |

