# OpenReview forum: "Visual Sparse Steering (VS2): Unsupervised Adaptation for Image Classification via Sparsity-Guided Steering Vectors"
_ICLR.cc/2026/Conference — Submitted to ICLR 2026_

### Official Review · Reviewer_fScH · 2025-10-26

**Soundness:** 2
**Presentation:** 3
**Contribution:** 2
**Rating:** 4
**Confidence:** 4

**Summary:**

The paper presents an unsupervised steering technique (VS2) to adapt vision models. The authors mainly test the approach on 3 common benchmarks using the CLIP ViTs backbones.

**Strengths:**

**S1.** The paper treats a relevant problem, i.e., adapting foundation models (FMs) for downstream classification tasks.

**S2.** The steering approach, as substantiated by the comparison shown in Table 1, appears to be more efficient than the baseline reconstruction task, making the idea interesting. I also find the analysis shown in Section 4.3 (Table 2) insightful.
From these findings, it indeed appears that steering removes some ambiguity among samples of different classes that share common features.

**S3.** The paper reads well, it is quite systematic in terms of analysis, and presents results and figures in a clear manner.

**Weaknesses:**

**W1.** Lacking related work: In my opinion, the authors do not perform a good job in locating the problem that they are trying to solve, and mostly focus on a particular technique derived from the natural language literature, but in practice are solving an unsupervised domain adaptation (UDA) problem: the encoder is a frozen generalist model (CLIP) that must be adapted to a particular classification task without using any labels. The current related work section does not mention any literature from UDA, despite its strong relevance.

**W2.** Missing unsupervised adaptation baselines: The paper, which primarily tests the approach on a UDA problem for pre-trained FMs, lacks a comparison against popular baselines such as test-time augmentation (TTA) (fully training-free).

**W3.** Missing unsupervised adaptation datasets: Typically, test-time adaptation addresses distribution shifts. CIFAR and TinyImagenet are surely ID for CLIP pre-training data. CUB could arguably be defined to be OOD since it contains several fine-grained bird species.

**W4.** Missing quantification of “lightweight” adaptation: The paper claims that the adaptation is lightweight, but it lacks proper comparison and quantification to support this claim.

**W5.** Missing reliability implementation: The possibility of checking the reconstruction loss to assess the “reliability” of the prediction in cases of OOD samples makes sense (Section 4.4). However, the authors, if I understand correctly, did not implement this mechanism in practice, making the claim somewhat vague and untested.

**W6.** Writing style: while asking questions usually raises the reader's interest, using them too frequently, as is done especially in the introduction, may end up sounding verbose and repetitive. This weakness is minor.

**Questions:**

**Q1.** What are the differences between your approach and the UDA literature? An additional related work paragraph would clarify this point.

**Q2.** Does VS2 work better than TTA? The latter does not need any training.

**Q3.** Does VS2 also work well for datasets that are more OOD than the tested ones (they are arguably ID)?

**Q4.** How lightweight is VS2 compared to full-finetuning/LORA adaptation? Clearly, you are not using labels, but in terms of FLOPS. I also didn’t manage to find specifics regarding the size of the SAE (possibly missed this detail).

**Q5.** How would you actually implement the reliability check using the reconstruction loss? This would likely add an additional hyperparameter. Would this prevent performance loss (e.g., see CUB)?

In summary, I found the paper interesting, but in its current form, it is more suitable for a workshop than a conference. Welcoming feedback from the authors during the discussion period.

---

> ### Author Response · Authors · 2025-11-25
>
> We would like to thank reviewer **fScH** for the thorough review. We are pleased you found our `“idea interesting”` and `“the analysis in Section 4.3 insightful,”` which `“indeed appears that steering removes some ambiguity among samples of different classes that share common features.”` We also appreciate your comments that the paper `“reads well,”` is `“quite systematic in terms of analysis,”` and `“presents results and figures in clear manner.”` We appreciate your comments, and we would like to provide you with further insights:
>
> ### Regarding Unsupervised Domain Adaptation (UDA) literature and Baselines
>
>
> > We would first like to (1) clarify our rationale and motivation, and (2) present additional insights and comparisons with baselines from the Test-Time Adaptation (TTA) literature, in order to further position our work within this domain.
>
> > The main purpose of our work is to showcase to the *mechanistic interpretability* community that **Sparse Autoencoders (SAEs) can be leveraged not only for interpretability, but also to improve downstream classification** performance. This is an *underexplored* direction within the community, and our intention was to position the paper accordingly. That said, we do appreciate your insight regarding *comparisons and positioning with respect to the Unsupervised Domain Adaptation (UDA) literature*. While not a contribution of our work, an important distinction of our approach is that **VS2 is inherently interpretable**, a property we think is valuable. We provide supporting evidence for this in Appendix A.2.
>
> > While interpretability is not a focus of most existing UDA or Test-Time Adaptation (TTA) literature, *we agree that providing comparisons with representative TTA methods is essential*. Accordingly, for the datasets in Table 1 of our main paper, we include baselines from TPT [1] , C-TPT [2], and Diff-TPT [3]. These methods optimize text prompts during adaptation, whereas our method does not modify or update the text prompts. In our Baseline setting, we use the fixed prompt “a photo of a {label}.” The Ensemble setting corresponds to a set of 80 fixed ImageNet-style templates, following CLIP’s zero-shot evaluation setup [4]. Finally, Aug refers to the test-time image augmentations used in TTA baselines. For efficiency, we apply only 10 image augmentations, compared to the 63 used in other methods.
>
> |  | CIFAR-100 | CUB-200 | Tiny-Imagenet |
> | --- | --- | --- | --- |
> | Baseline | 61.07 | 51.76 | 56.64 |
> | Ensemble [4] | 63.66 | 51.54 | 61.39 |
> | TPT [1]  | 64.09 | 51.83 | 62.77 |
> | C-TPT [2] | `64.86` | 52.54 | **`63.20`** |
> | Diff-TPT [3] | 63.04 | **`52.80`** | 61.60 |
> | VS2 (ours) | 64.52 | `52.69` | 58.14 |
> | VS2 (ours) + Ensemble [4] | **`65.48`** | 52.47 | `62.79` |
> | VS2 (ours) + Ensemble [4] + Aug | 65.80 | 51.54 | 62.59 |
>
> > We observe that our method *performs comparably to existing Test-Time Adaptation techniques*, achieving the **best accuracy on CIFAR-100** and **second-best performance on CUB-200 and Tiny ImageNet**. However, a *key advantage* of our approach is its *inherent interpretability*, an aspect not present in the compared methods. Moreover, our method is potentially modality-agnostic and could be extended to both language and vision domains.
>
> [1] Shu et al. Test-Time Prompt Tuning for Zero-Shot Generalization in Vision-Language Models. 2022.
>
> [2] Yoon et al. C-TPT Calibrated Test-Time Prompt Tuning for Vision-Language Models via Text Feature Dispersion. 2024.
>
> [3] Feng et al. Diverse Data Augmentation with Diffusions for Effective Test-Time Prompt Tuning. 2023.
>
> [4] Radford et al. Learning Transferable Visual Models from Natural Language Supervision. 2021.

---

> > ### Author Response · Authors · 2025-11-25
> >
> > ### Extending the Evaluation of VS2 to Additional Datasets
> >
> > > This is an interesting question, and we agree that **evaluating on more TTA datasets strengthens our findings**. To this end, we apply the same methodology used in the main paper to **six additional datasets**: Aircraft [5], Food101 [6], Flower102 [7], Caltech101 [8], SUN [9], and EuroSAT [10]. For classification, we use standard CLIP text prompts of the form “a photo of a {label}” (Baseline), and Ens, which refers to an ensemble of 80 ImageNet-style templates.
> >
> > | Method / Dataset | Aircraft | Food101 | Flower102 | Caltech101 | SUN | EuroSAT |
> > | --- | --- | --- | --- | --- | --- | --- |
> > | **Baseline** | 24.84 | 80.71 | 63.72 | 78.53 | 51.79 | 31.61 |
> > | **VS2** | 27.69 | 81.20 | 64.81 | 80.95 | 54.80 | 34.65 |
> > | **VS2 Gain** | **+2.85** | **+0.49** | **+1.09** | **+2.42** | **+3.01** | **+3.04** |
> > |  |  |  |  |  |  |  |
> > | **Baseline + Ensemble** | 26.55 | 81.90 | 63.33 | 77.32 | 57.79 | 33.98 |
> > | **VS2 + Ensemble** | 28.80 | 82.06 | 63.77 | 80.43 | 60.46 | 36.80 |
> > | **VS2+Ens Gain** | **+2.25** | **+0.16** | **+0.44** | **+3.11** | **+2.67** | **+2.82** |
> >
> > > We observe that in-domain SAEs provide **significant contributions in all 6 new dataset of up to +3.04%**.
> >
> > [5] Maji et al. Fine-Grained Visual Classification of Aircraft. 2013.
> >
> > [6] Bossard et al. Food-101 - Mining Discriminative Components with Random Forests. 2014.
> >
> > [7] Nilsback et al. Automated Flower Classification over a Large Number of Classes. 2008.
> >
> > [8] Fei-Fei et al. Learning Generative Visual Models from Few Training Examples: An Incremental Bayesian Approach Tested on 101 Object Categories. 2004.
> >
> > [9] Xiao et al. SUN Database: Large-Scale Scene Recognition from Abbey to Zoo. 2010.
> >
> > [10] Helber et al. EuroSAT: A Novel Dataset and Deep Learning Benchmark for Land Use and Land Cover Classification. 2017.

---

> > > ### Author Response · Authors · 2025-11-25
> > >
> > > ### On the VS2 Fallback Mechanism
> > >
> > > > In Table 3 of the main paper, we provide evidence that **high FVU values (specifically, FVU > 1) are associated with performance degradation**. It is important to clarify that *FVU is a dataset-level metric*, computed using the mean squared reconstruction error and the variance of the original data. If test samples were processed in sufficiently large batches, FVU could still be computed meaningfully. However, in this section, we focus on per-instance analysis to investigate how a simple reconstruction error metric could be used for decision-making at inference time. In the per-instance setting, we cannot compute global means and variances, but we can use the *raw reconstruction loss as a proxy*. Our goal is to assess how downstream classification accuracy changes as we vary the reconstruction loss threshold, thereby evaluating the effectiveness of this signal for fallback decisions.
> > >
> > > > We would like to provide additional evidence of how the fallback mechanism operates in practice. Specifically, we use the "Generalized SAE" from Table 3, trained on CIFAR-100, CUB-200, and Tiny ImageNet, and apply steering to the new datasets discussed earlier, which *the SAE has not seen during training*. As these datasets are out-of-distribution, we do not expect the generalized SAE to perform well across all cases, and we anticipate failures in certain domains. To mitigate such failures, we introduce a Fallback Mechanism that activates based on the reconstruction loss of each test sample. This mechanism involves a **single tunable hyperparameter: a reconstruction threshold**. When a test image is passed through the SAE, we compute its reconstruction loss. If the loss is below the threshold, we apply steering; otherwise, we revert to the original model prediction. In the following experiment, we report the results of this selective steering process and analyze how performance changes under different thresholds.
> > > |  | Threshold | Aircraft | Food101 | Flower102 | Caltech101 | Sun | EuroSAT |
> > > | --- | --- | --- | --- | --- | --- | --- | --- |
> > > | **Baseline** | — | 24.84 | 80.71 | 63.72 | 78.53 | 51.79 | 31.61 |
> > > | **VS2 (In-domain SAE)** | — | 27.69 (**+2.85**) | 81.20 (**+0.49**) | 64.81 (**+1.09**) | 80.95 (**+2.42**) | 54.80 (**+3.01**) | 34.65 (**+3.04**) |
> > > | **VS2 (Generalized SAE)** | — | 18.57 (**−6.27**) | 81.06 (**+0.35**) | 64.11 (**+0.39**) | 79.72 (**+1.19**) | 45.78 (**−6.01**) | 30.76 (**−0.85**) |
> > > | **VS2 (Generalized) + Fallback** | **1.5** | 24.87 (**+0.03**) | 80.72 (**+0.01**) | 63.70 (**−0.02**) | 78.53 (**+0.00**) | 51.79 (**+0.00**) | 31.61 (**+0.00**) |
> > > |  | **2.0** | 24.87 (**+0.03**) | 80.72 (**+0.01**) | 63.70 (**−0.02**) | 78.53 (**+0.00**) | 51.79 (**+0.00**) | 31.61 (**+0.00**) |
> > > |  | **2.5** | 24.87 (**+0.03**) | 80.72 (**+0.01**) | 63.70 (**−0.02**) | 78.53 (**+0.00**) | 51.79 (**+0.00**) | 31.61 (**+0.00**) |
> > > |  | **3.0** | 24.87 (**+0.03**) | 80.72 (**+0.01**) | 63.70 (**−0.02**) | 78.53 (**+0.00**) | 51.79 (**+0.00**) | 31.61 (**+0.00**) |
> > > |  | **3.5** | 24.87 (**+0.03**) | 80.72 (**+0.01**) | 63.70 (**−0.02**) | 78.51 (**−0.02**) | 51.79 (**+0.00**) | 30.11 (**−1.50**) |
> > > |  | **4.0** | 24.87 (**+0.03**) | 80.71 (**+0.00**) | 63.88 (**+0.16**) | 78.11 (**−0.42**) | 51.79 (**+0.00**) | 30.41 (**−1.20**) |
> > > |  | **4.5** | 24.87 (**+0.03**) | 80.69 (**−0.02**) | 64.19 (**+0.47**) | 78.42 (**−0.11**) | 51.46 (**−0.33**) | 30.76 (**−0.85**) |
> > > |  | **5.0** | 24.12 (**−0.72**) | 80.81 (**+0.10**) | 64.16 (**+0.44**) | 79.39 (**+0.86**) | 49.11 (**−2.68**) | 30.76 (**−0.85**) |
> > > |  | **5.5** | 21.81 (**−3.03**) | 81.03 (**+0.32**) | 64.08 (**+0.36**) | 79.79 (**+1.26**) | 46.82 (**−4.97**) | 30.76 (**−0.85**) |
> > > |  | **6.0** | 18.93 (**−5.91**) | 81.05 (**+0.34**) | 64.10 (**+0.38**) | 79.70 (**+1.17**) | 46.10 (**−5.69**) | 30.76 (**−0.85**) |
> > > |  | **6.5** | 18.69 (**−6.15**) | 81.05 (**+0.34**) | 64.11 (**+0.39**) | 79.72 (**+1.19**) | 45.86 (**−5.93**) | 30.76 (**−0.85**) |
> > > |  | **7.0** | 18.57 (**−6.27**) | 81.06 (**+0.35**) | 64.11 (**+0.39**) | 79.72 (**+1.19**) | 45.79 (**−6.00**) | 30.76 (**−0.85**) |
> > >
> > > > We observe that the *“Generalized SAE” yields **accuracy gains** on Food101, Flower102, and Caltech101, but leads to **notable drops** on SUN and Aircraft*. **These failures, however, are effectively mitigated by the per-instance Fallback Mechanism**, which dynamically chooses whether to apply steering based on reconstruction loss. This result **highlights the practical importance of adaptive strategies for robust generalization, especially when deploying SAEs on unseen or out-of-distribution datasets**.

---

> > > > ### Author Response · Authors · 2025-11-25
> > > >
> > > > ### Computational Overhead: Lightweight Claim
> > > >
> > > > > To clarify the **computational overhead** of our method, we report the exact <u>inference</u> **FLOPs**, **MACs**, and **parameter counts** for the CLIP ViT-B/32 vision encoder under the baseline model, VS2, and LoRA.
> > > >
> > > >
> > > > | Method | GFLOPs (fwd) | GMACs (fwd) | Params (M) |
> > > > | --- | --- | --- | --- |
> > > > | Plain CLIP (vision) | **8.7295** | **4.3623** | **87.456** |
> > > > | VS2 (SAE steering) | **8.7342** | **4.3647** | **92.178** |
> > > > | LoRA (rank = 16) | **8.7885** | **4.3918** | **88.046** |
> > > >
> > > > > VS2 **adds only 0.0047 GFLOPs over the baseline**, less than 0.1%, because it applies a lightweight SAE operation to the *CLS token in a single layer*, whereas LoRA introduces additional projections inside every attention block across all tokens and layers, leading to noticeably higher FLOPs despite having fewer parameters.
> > > >
> > > > > To further quantify the cost of adapting CLIP, we report per-sample <u>training</u> FLOPs and trainable parameters for SAE (with and without cached activations), LoRA, and full fine-tuning on the ViT-B/32 vision encoder.
> > > >
> > > > | Method | Training FLOPs (fwd + bwd) | FLOPs (G) | Trainable Params |
> > > > | --- | --- | --- | --- |
> > > > | SAE (cached activations) | 14,164,992 | **0.014** | 4,722,432 |
> > > > | SAE (non-cached; incl. extraction) | 8,743,645,440 | **8.744** | 4,722,432 |
> > > > | LoRA (r = 16) | 26,365,388,544 | **26.365** | 589,824 |
> > > > | Full Fine-tuning (vision encoder) | 26,188,441,344 | **26.188** | 87,456,000 |
> > > >
> > > > > Even when including activation extraction, SAE remains over around 3× cheaper than LoRA or full fine-tuning, and when cached activations are used, as in VS2, the training cost drops to only 0.014 GFLOPs per sample.
> > > >
> > > > > Finally, **Test-Time Adaptation (TTA) methods**, such as TPT and its variants, introduce **substantial computational overhead**. This is primarily due to the use of *multiple image augmentations*, *repeated test-time optimization steps*, and the *requirement to backpropagate gradients to update prompts during inference*. A plain CLIP ViT-B/32 forward pass on a single 224×224 image, together with 100 text prompts (one for each class given that the number of labels is 100), costs 590.6 GFLOPs in total (8.7 GFLOPs for the vision encoder and 581.8 GFLOPs for the text encoder). In contrast, in our current testbed, TPT performs test-time adaptation by generating 63 augmented views of the test image and running 4 optimization steps, each of which requires a full CLIP forward pass over all views plus a backward pass through the text encoder. This results in 9.217 TFLOPs of adaptation cost per test image, followed by a final CLIP inference of 0.591 TFLOPs, for a total of 9.807 TFLOPs. Overall, with these computations, TPT requires 16.6× more computation per image than plain CLIP. Importantly, this overhead is incurred at inference time, making TPT far more expensive than VS2.
> > > >
> > > > > **Regarding SAE size**: The SAE consists of a learned encoder $E \in \mathbb{R}^{768 \times 3072} $ and decoder $D \in \mathbb{R}^{3072 \times 768}$. We provide details in Appendix C. The expansion factor of 4 indicates that the sparse feature dimensionality is 4× larger than the original embedding size, enabling a richer sparse representation while maintaining reasonable inference overhead. Notably, the by-definition sparsity (e.g., using top-64 active features with the rest set to zero) also helps minimize computational overhead during decoding. Specifically, when the sparse latent vector \( z \) is multiplied with the decoder \( D \), we only need to compute the contributions from the 64 non-zero features, avoiding unnecessary multiplications with zeros.

---

> > > > > ### Comment · Reviewer_fScH · 2025-11-26
> > > > >
> > > > > I would like to thank the authors for their efforts in providing a detailed rebuttal. Please find my comments and answers below.
> > > > >
> > > > > **W1.** I understood the point of view of the authors and thank them for clarifying it. The paper should, however, include a related work paragraph that situates the work in relation to established baselines, because the evaluations performed are primarily for UDA.
> > > > >
> > > > > **W2.** The comparison against TTA baselines demonstrates that VS2 aligns in terms of performance. To me, it then seems that possible advantages could be "interpretability" (the authors need to link an analysis that proves this point) and "efficiency" (see rebuttal for **W4**). It is critical to explain these advantages in the main manuscript.
> > > > >
> > > > > **W3.** The testing on more OOD datasets (e.g., EuroSAT) proves that VS2 generalizes to more OOD scenarios. This analysis would clearly strengthen the paper.
> > > > >
> > > > > **W4.** The quantification of "lightweight" is addressed.
> > > > >
> > > > > **W5.** I think that this new thresholding approach complements well the FVU analysis, which currently does not have practical benefits. From the table, it appears that a tight threshold (1.5/2) can avoid performance degradation compared to the baseline, which is beneficial. The in-domain SAE is still clearly on top in those cases. I also read the answer of the Reviewer **qejr** to this point and tend to understand their perspective. To me, however, it would be sufficient that the authors acknowledge this as a clear limitation and explain that for OOD datasets, the generalized SAE might default to the baseline (as practically demonstrated by the thresholding fallback approach), and future work would be needed. It is critical that this new thresholding analysis is included in the main manuscript.
> > > > >
> > > > > In summary, most of my concerns have been addressed by the rebuttal here on OpenReview. That being said, I would like to see the updated main manuscript with the above analyses included, as it would, in my opinion, elevate the paper from marginally below to marginally above the acceptance threshold.

---

> > > > > > ### Author Response · Authors · 2025-11-27
> > > > > >
> > > > > > We want to thank Reviewer **fScH** for the valuable feedback, which helped us improve the quality of the paper. We are glad the reviewer acknowledged our `“efforts in providing a detailed rebuttal”` and that `“most of [their] concerns have been addressed by the rebuttal.”` We have updated the main manuscript accordingly (highlighted in blue). Specifically:
> > > > > >
> > > > > > **W1.** We have included a paragraph on **related work** in `Lines 162–175`, to better position our contributions within the context of prior literature in UDA, and more specifically, Test-Time Adaptation.
> > > > > >
> > > > > > **W2.** We incorporated the **TTA baselines** into the main manuscript (`Lines 352–363`) and `Table 2a`, emphasizing the advantages of VS2 in terms of both *interpretability* and *inference-time computational efficiency*.
> > > > > >
> > > > > > **W3.** We reported the performance of **VS2 across all six new datasets** (including EuroSAT) in `Table 4`, providing a more complete view of its generalization behavior.
> > > > > >
> > > > > > **W4.** We quantified the **computational efficiency** of VS2 in `Lines 365–377` of the main manuscript and further detailed it in `Appendix N`.
> > > > > >
> > > > > > **W5.** We introduced the **Fallback Mechanism** in `Lines 475–485` of the main manuscript and provided additional details in `Appendix O`. We explicitly stated that *“the fallback mechanism primarily serves as a safety measure, defaulting to the baseline on out-of-distribution (OOD) datasets when necessary,* while also highlighting the development of more nuanced reliability mechanisms as a promising direction for *future work*.
> > > > > >
> > > > > > **Additional Change**: Sensitivity to sparse amplification ($\gamma$) and steering magnitude ($\lambda$) is now primarily discussed in the appendix. The corresponding figure (previously in the main paper), Figure 3, has been moved to Appendix B to conserve space in the main manuscript.
> > > > > >
> > > > > > Finally, we would like to thank Reviewer fScH once again, and we welcome any further comments or suggestions that could help provide additional clarification and further improve our work.

---

> > > > > > > ### Comment · Reviewer_fScH · 2025-11-28
> > > > > > >
> > > > > > > I thank the authors for having updated the manuscript. I will accordingly update the review and score from 4 to 6 once the authors' response period ends.

---

> > > > > > > > ### Author Response · Authors · 2025-12-01
> > > > > > > >
> > > > > > > > We would like to thank reviewer fScH for the thorough review and the insightful questions, which helped clarify and strengthen our paper. We are pleased to hear that you will update the review and score from 4 to 6 once the authors' response period ends. Should you have any further questions or require additional clarifications, we would be pleased to provide further insights.

---

### Official Review · Reviewer_qejr · 2025-10-31

**Soundness:** 2
**Presentation:** 3
**Contribution:** 2
**Rating:** 4
**Confidence:** 4

**Summary:**

This paper presents Visual Sparse Steering (VS2) and its retrieval-augmented variant, VS2++, to steer vision foundation models at test time. The former method, VS2, constructs a steering vector by upweighting sparse features extracted from a sparse autoencoder (SAE) trained on the model's internal activations. This label-free, test-time method is empirically shown to improve CLIP zero-shot classification across three different datasets. Furthermore, in a relaxed setting where retrieval from an external unlabeled corpus is allowed, the proposed VS2++ constructs a steering vector from contrastive examples built using pseudo-labels, resulting in a large boost in classification accuracy. The authors also claim that VS2 includes a built-in reliability diagnostic which is not present in common steering vectors: the SAE's reconstruction loss is shown to correlate with classification performance.

**Strengths:**

- **Novel Approach:** The proposed VS2 is interesting, as steering vectors are constructed by simply upweighting all sparse features but results in performance gain. Unlike conventional steering vectors, VS2 is label-free method avoiding the need for contrastive data.
- **Consistent Empirical Improvements:** The proposed VS2++ method demonstrates consistent and notable empirical improvements over the strong CLIP zero-shot baseline across all three evaluated datasets (CIFAR-100, CUB-200, and Tiny-ImageNet).
- **Practical Reliability Diagnostic:** The paper also includes discussion about having reliability diagnostic, an overlooked but important issue in practical applications.

**Weaknesses:**

- **Overclaimed Scope of Contribution:** The paper's central claim is to "steer vision foundation models", but the empirical validation is exclusively limited to improving the zero-shot classification accuracy of CLIP models. There is no evidence provided that this method can generalize to other vision foundation models or other tasks beyond classification. This discrepancy makes the general claim of "steering" appear to be an overclaim. The method is more accurately framed as a test-time adaptation technique for CLIP zero-shot classification.
- **Misleading "Test-Time" Claim and Practicality Issues:** The method is presented as a "test-time" approach, but this is a bit misleading. Its effectiveness hinges on a pretrained SAE that must be trained on the in-domain activations of the target dataset's training split. This requirement has several critical weaknesses:
    - The method cannot work as a "plug-and-play" method, as it requires substantial, dataset-specific computation and full access to the (unlabeled) training set before any test sample can be processed.
    - The authors' own experiment with a "generalized SAE" (trained on the union of datasets) in Section 4.4 confirms this limitation, as it fails to generalize and underperforms the baseline on CUB-200. This reinforces the method's strong dependency on in-domain data.
    - This "pre-training" step on the training set blurs the line between a true test-time method and a form of unsupervised domain adaptation.
- **Contradictory Premise for Data-Scarce Scenarios:** The chosen experiment setup, CLIP zero-shot classification, is most valuable in data-scarce (e.g., few-shot or long-tail) domains where training data is limited. However, these are precisely the scenarios where VS2/VS2++ would be least applicable.
    - VS2 would fail because there is insufficient in-domain data to train a reliable SAE.
    - VS2++ would be even more likely to fail, as it requires a large high-quality retrieval corpus, which is unlikely to exist in a data-scarce setting.
- **Unvalidated Reliability Diagnostic:** The "built-in reliability diagnostic" is presented as a key advantage but is not empirically validated as a practical tool. The analysis in Section 4.4 and Table 3 only demonstrates a dataset-level correlation (i.e., CUB-200 has high average FVU and low accuracy). It fails to provide evidence that this works on a per-instance basis, which is necessary for it to function as a useful fallback mechanism.

**Questions:**

- **Generalizability:** Given the claim of "steering vision foundation models," could the authors elaborate on a concrete way to apply VS2 to (a) non-CLIP models like DINOv2 and (b) non-classification tasks like VQA or image generation, where the concept of a single "CLS token" or "classification accuracy" may not directly apply?
- **Sensitivity to Data Scarcity:** How does the SAE's reconstruction error (and the final VS2 accuracy) degrade as the proportion of the training set used for SAE training is reduced? Would VS2 work well under data-scarce setting where CLIP zero shot classification is often considered?
- **Missing Experimental Details:** Could the authors please clarify the setup for the "CLIP classifier" used for pseudo-labeling? Is this a standard zero-shot classifier? If so, what text prompts were used? These details are essential for reproducibility.
- **Validating the Fallback Mechanism:** To validate the *per-instance* utility of the reliability diagnostic, would it be possible to bin the test instances by their reconstruction loss? A plot showing accuracy per bin would confirm if instances with higher loss are indeed those where VS2 underperforms. Following this, have the authors experimented with implementing the actual fallback mechanism (i.e., if loss > threshold, use the baseline prediction)? This would show if the diagnostic practically improves the final accuracy.
- **More Baselines for VS2++:** For the VS2++ setting, where access to the unlabeled training set is assumed, there could be simpler but potentially stronger baselines. How does VS2++ compare against:
    - (a) A simple k-NN classifier using retrieval (i.e., a majority vote of the pseudo-labels from the top-k retrieved neighbors)?
    - (b) A lightweight linear probe trained on the training set's CLS tokens and their corresponding pseudo-labels (or in oracle setting, labels)?

---

> ### Author Response · Authors · 2025-11-25
>
> We would like to thank reviewer **qejr** for the thorough review and the insightful comments. We are glad you found our approach `“interesting”` and `“novel,”` with `“consistent empirical improvements,”` and that you highlighted the `“practical reliability diagnostic”` of our method as an `“overlooked but important issue in practical applications.”` We greatly appreciate your feedback and would like to provide additional results and clarifications to further support our contributions.
>
>
> ### Evaluating Out-of-Distribution Generalization
>
> > We would first like to (1) clarify the rationale and motivation behind our experiments, and (2) present additional insights on more datasets using the Generalized SAE introduced in Section 4.4.
>
> #### 1. Clarification on Rationale of Work and Contribution
>
> > The main purpose of our work is to demonstrate to the *mechanistic interpretability* community that **Sparse Autoencoders (SAEs) can be leveraged not only for interpretability, but also to improve downstream classification performance**. Training effective SAEs remains an active and largely open research direction [1, 2]. To evaluate the potential of our approach, we begin with the assumption: *given a well-trained SAE, can it enhance performance?* We *define a “good” SAE as one trained on in-domain data*, and we show that, indeed, using such an SAE can lead to **measurable improvements in downstream accuracy**.
>
> > We, then,  aim to train a **more general SAE** by incorporating all training datasets during training and evaluating its utility on downstream tasks. While the resulting SAE remains in-domain, we observe that although it maintains strong performance on CIFAR-100 and Tiny-ImageNet, it leads to degraded performance on CUB, as you identified. Although CUB-200 is part of the in-domain training data, it still performs poorly, likely due to suboptimal SAE training caused by dataset imbalance (e.g., ~50K samples from CIFAR-100, ~12K from CUB-200, and ~100K from Tiny ImageNet).
>
> #### 2. VS2 on unseen datasets using a Generalized SAE
>
> > To further extend our contribution to **additional datasets**, we include results on Aircraft [3], Food101 [4], Flower102 [5], Caltech101 [6], SUN [7], and EuroSAT [8]. To highlight the impact of SAE training regimes, we first report results using SAEs trained in-domain for each dataset. We then compare these with the performance of the Generalized SAE trained jointly on CIFAR-100, CUB-200, and Tiny ImageNet. For classification, we follow standard CLIP evaluation protocols using the prompt “a photo of a {label}” for the Baseline, and Ens, which consists of the 80 ImageNet-style prompt templates [9].
>
> | Method / Dataset | Aircraft | Food101 | Flower102 | Caltech101 | SUN | EuroSAT |
> | --- | --- | --- | --- | --- | --- | --- |
> | **Baseline** | 24.84 | 80.71 | 63.72 | 78.53 | 51.79 | 31.61 |
> | **VS2** | 27.69 | 81.20 | 64.81 | 80.95 | 54.80 | 34.65 |
> | **VS2 Gain** | **+2.85** | **+0.49** | **+1.09** | **+2.42** | **+3.01** | **+3.04** |
> |  |  |  |  |  |  |  |
> | **Baseline + Ensemble [9]** | 26.55 | 81.90 | 63.33 | 77.32 | 57.79 | 33.98 |
> | **VS2 + Ensemble [9]** | 28.80 | 82.06 | 63.77 | 80.43 | 60.46 | 36.80 |
> | **VS2+Ens Gain** | **+2.25** | **+0.16** | **+0.44** | **+3.11** | **+2.67** | **+2.82** |
>
> We observe that **in-domain SAEs yield significant performance improvements on most datasets, up to +3.04% in some cases**. In the table below, we report the performance of *VS2 on datasets that were not part of the SAE’s training distribution*.
>
> |  | Aircraft | Food101 | Flower102 | Caltech101 | Sun | Eurosat |
> | --- | --- | --- | --- | --- | --- | --- |
> | Baseline | 24.84 | 80.71 | 63.72 | 78.53 | 51.79 | 31.61 |
> | VS2 (In-domain SAE) | 27.69 (**+2.85)** | 81.20 (**+0.49)** | 64.81 (+1.09) | 80.95 (**+2.42)** | 54.80 (**+3.01)** | 34.65 (**+3.04)** |
> | VS2 (Generalized SAE) | 18.57 (**−6.27)** | 81.06 (**+0.35)** | 64.11 (**+0.39)** | 79.72 (**+1.19)** | 45.78 (**−6.01)** | 30.76 (**−0.85)** |
>
> > The Generalized SAE, when evaluated on entirely out-of-domain datasets, achieves **accuracy gains on Food101, Flower102, and Caltech101; shows a moderate drop on EuroSAT; and suffers significant performance degradation on Aircraft and SUN**. These latter two cases, Aircraft and SUN, are **precisely where a fallback mechanism becomes essential, as discussed in the following section**, by *mitigating harmful steering* when the SAE reconstruction signal indicates a poor fit.

---

> > ### Author Response · Authors · 2025-11-25
> >
> > ### On the VS2 Fallback Mechanism
> >
> > > In Table 3 of the main paper, we provide evidence that **high FVU values (specifically, FVU > 1) are associated with performance degradation**. It is important to clarify that *FVU is a dataset-level metric*, computed using the mean squared reconstruction error and the variance of the original data. If test samples were processed in sufficiently large batches, FVU could still be computed meaningfully. However, in this section, we focus on per-instance analysis to investigate how a simple reconstruction error metric could be used for decision-making at inference time. In the per-instance setting, we cannot compute global means and variances, but we can use the *raw reconstruction loss as a proxy*. Our goal is to assess how downstream classification accuracy changes as we vary the reconstruction loss threshold, thereby evaluating the effectiveness of this signal for fallback decisions.
> >
> > > We would like to provide additional evidence of how the fallback mechanism operates in practice. Specifically, we use the "Generalized SAE" from Table 3, trained on CIFAR-100, CUB-200, and Tiny ImageNet, and apply steering to the new datasets discussed earlier, which *the SAE has not seen during training*. As these datasets are out-of-distribution, we do not expect the generalized SAE to perform well across all cases, and we anticipate failures in certain domains. To mitigate such failures, we introduce a Fallback Mechanism that activates based on the reconstruction loss of each test sample. This mechanism involves a **single tunable hyperparameter: a reconstruction threshold**. When a test image is passed through the SAE, we compute its reconstruction loss. If the loss is below the threshold, we apply steering; otherwise, we revert to the original model prediction. In the following experiment, we report the results of this selective steering process and analyze how performance changes under different thresholds.
> > |  | Threshold | Aircraft | Food101 | Flower102 | Caltech101 | Sun | EuroSAT |
> > | --- | --- | --- | --- | --- | --- | --- | --- |
> > | **Baseline** | — | 24.84 | 80.71 | 63.72 | 78.53 | 51.79 | 31.61 |
> > | **VS2 (In-domain SAE)** | — | 27.69 (**+2.85**) | 81.20 (**+0.49**) | 64.81 (**+1.09**) | 80.95 (**+2.42**) | 54.80 (**+3.01**) | 34.65 (**+3.04**) |
> > | **VS2 (Generalized SAE)** | — | 18.57 (**−6.27**) | 81.06 (**+0.35**) | 64.11 (**+0.39**) | 79.72 (**+1.19**) | 45.78 (**−6.01**) | 30.76 (**−0.85**) |
> > | **VS2 (Generalized) + Fallback** | **1.5** | 24.87 (**+0.03**) | 80.72 (**+0.01**) | 63.70 (**−0.02**) | 78.53 (**+0.00**) | 51.79 (**+0.00**) | 31.61 (**+0.00**) |
> > |  | **2.0** | 24.87 (**+0.03**) | 80.72 (**+0.01**) | 63.70 (**−0.02**) | 78.53 (**+0.00**) | 51.79 (**+0.00**) | 31.61 (**+0.00**) |
> > |  | **2.5** | 24.87 (**+0.03**) | 80.72 (**+0.01**) | 63.70 (**−0.02**) | 78.53 (**+0.00**) | 51.79 (**+0.00**) | 31.61 (**+0.00**) |
> > |  | **3.0** | 24.87 (**+0.03**) | 80.72 (**+0.01**) | 63.70 (**−0.02**) | 78.53 (**+0.00**) | 51.79 (**+0.00**) | 31.61 (**+0.00**) |
> > |  | **3.5** | 24.87 (**+0.03**) | 80.72 (**+0.01**) | 63.70 (**−0.02**) | 78.51 (**−0.02**) | 51.79 (**+0.00**) | 30.11 (**−1.50**) |
> > |  | **4.0** | 24.87 (**+0.03**) | 80.71 (**+0.00**) | 63.88 (**+0.16**) | 78.11 (**−0.42**) | 51.79 (**+0.00**) | 30.41 (**−1.20**) |
> > |  | **4.5** | 24.87 (**+0.03**) | 80.69 (**−0.02**) | 64.19 (**+0.47**) | 78.42 (**−0.11**) | 51.46 (**−0.33**) | 30.76 (**−0.85**) |
> > |  | **5.0** | 24.12 (**−0.72**) | 80.81 (**+0.10**) | 64.16 (**+0.44**) | 79.39 (**+0.86**) | 49.11 (**−2.68**) | 30.76 (**−0.85**) |
> > |  | **5.5** | 21.81 (**−3.03**) | 81.03 (**+0.32**) | 64.08 (**+0.36**) | 79.79 (**+1.26**) | 46.82 (**−4.97**) | 30.76 (**−0.85**) |
> > |  | **6.0** | 18.93 (**−5.91**) | 81.05 (**+0.34**) | 64.10 (**+0.38**) | 79.70 (**+1.17**) | 46.10 (**−5.69**) | 30.76 (**−0.85**) |
> > |  | **6.5** | 18.69 (**−6.15**) | 81.05 (**+0.34**) | 64.11 (**+0.39**) | 79.72 (**+1.19**) | 45.86 (**−5.93**) | 30.76 (**−0.85**) |
> > |  | **7.0** | 18.57 (**−6.27**) | 81.06 (**+0.35**) | 64.11 (**+0.39**) | 79.72 (**+1.19**) | 45.79 (**−6.00**) | 30.76 (**−0.85**) |
> >
> > > We observe that the *“Generalized SAE” yields **accuracy gains** on Food101, Flower102, and Caltech101, but leads to **notable drops** on SUN and Aircraft*. **These failures, however, are effectively mitigated by the per-instance Fallback Mechanism**, which dynamically chooses whether to apply steering based on reconstruction loss. This result **highlights the practical importance of adaptive strategies for robust generalization, especially when deploying SAEs on unseen or out-of-distribution datasets**.

---

> > > ### Author Response · Authors · 2025-11-25
> > >
> > > ### On the Generalizability of the Term “Steering Vision Foundation Models”
> > >
> > > > We agree with the reviewer that our method is focused on **Vision Transformers** in the context of **CLIP-based zero-shot classification**. We avoided the term “zero-shot” as our approach involves training the SAE on **unlabeled data**, albeit with the assumption, in our initial experiments, that a well-trained SAE is available. To this end, we positioned the paper as “unsupervised adaptation”. We welcome the suggestion and are open to making changes that will further strengthen and better position the paper.
> > >
> > > > We use **CLIP** as our foundation because it *enables evaluation in a fully unlabeled setting, which aligns with our focus on unsupervised adaptation*. While our main focus is on downstream classification, the broader methodology is not limited to Vision Transformers or [CLS] token representations. For other architectures, including those without [CLS] tokens or even closed-source models, one could train SAEs on the final pooled embeddings produced by the model. Beyond classification, similar ideas have begun to emerge in related domains: for example, SAEs have been applied to concept unlearning in diffusion models [10] and multimodal alignment in large language–vision models [11].
> > >
> > >
> > >
> > > ### Experimental Details: CLIP Classifier
> > >
> > > > Regarding experimental details, the CLIP classifier used for pseudo-labeling is the standard zero-shot classifier, which employs the text prompt “a photo of a {label}.”
> > >
> > > ### Additional VS2++ Baselines
> > >
> > > > In the VS2++ setting, we evaluate against stronger baselines, as suggested. Specifically, we compare our sparse steering mechanism to a k-NN baseline using the final feature embeddings. The results are summarized below. We observe significant gains on CUB-200, comparable performance on CIFAR-100, and a performance drop on Tiny ImageNet, highlighting dataset-dependent effectiveness. This result further supports our observation in the paper, "highlighting the need for better feature selection under weakly supervised retrieval." Even in the oracle setting, where ground-truth labels are available, a significant accuracy gap remains, highlighting the potential for future contributions on how to selectively amplify sparse features.
> > >
> > > |  | Steering | k-NN |
> > > | --- | --- | --- |
> > > | CIFAR-100 | 77.11 | 77.38 |
> > > | CUB-200 | 52.81 | 46.44 |
> > > | TinyImagenet | 72.12 | 74.35 |
> > >
> > > > We note that linear probing would require access to labeled data, which falls outside our evaluation scope focused on unlabeled, zero-shot, unsupervised adaptation.
> > >
> > > ### Experimental Details and Data-scarce Scenarios
> > > > **Experimental Details.** Regarding experimental details, the CLIP classifier used for pseudo-labeling is the standard zero-shot classifier, which employs the text prompt “a photo of a {label}.”
> > >
> > > > **On data-scarce scenarios.** We agree that in data-scarce settings, training a reliable SAE can be challenging. In such cases, if the underlying data distribution is sufficiently captured by the SAE, we *still expect steering to yield performance gains*. Otherwise, *when the SAE fails to generalize*, our *fallback mechanism is designed to detect and mitigate these failures* by reverting to the original model.

---

> > > > ### Author Response · Authors · 2025-11-25
> > > >
> > > > ### References
> > > > [1] Gao et al. Scaling and Evaluating Sparse Autoencoders. ICLR 2025.
> > > >
> > > > [2] Bussmann et al. BatchTopK Sparse Autoencoders. 2024.
> > > >
> > > > [3] Maji et al. Fine-Grained Visual Classification of Aircraft. 2013.
> > > >
> > > > [4] Bossard et al. Food-101 - Mining Discriminative Components with Random Forests. 2014.
> > > >
> > > > [5] Nilsback et al. Automated Flower Classification over a Large Number of Classes. 2008.
> > > >
> > > > [6] Fei-Fei et al. Learning Generative Visual Models from Few Training Examples: An Incremental Bayesian Approach Tested on 101 Object Categories. 2004.
> > > >
> > > > [7] Xiao et al. SUN Database: Large-Scale Scene Recognition from Abbey to Zoo. 2010.
> > > >
> > > > [8] Helber et al. EuroSAT: A Novel Dataset and Deep Learning Benchmark for Land Use and Land Cover Classification. 2017.
> > > >
> > > > [9] Radford et al. Learning Transferable Visual Models from Natural Language Supervision. 2021.
> > > >
> > > > [10] Cywiński et al. SAeUron Interpretable Concept Unlearning in Diffusion Models with Sparse Autoencoders. 2025.
> > > >
> > > > [11] Lou et al. SAE-V Interpreting Multimodal Models for Enhanced Alignment. 2025.

---

> ### Comment · Reviewer_qejr · 2025-11-26
>
> I appreciate the authors' time and effort in providing clarifications and additional results. However, I still have several concerns and questions.
>
> ### **Presentation**
>
> The stated goal is "to demonstrate to the mechanistic interpretability community that Sparse Autoencoders (SAEs) can be leveraged not only for interpretability, but also to improve downstream classification performance." In its current form, the manuscript does not clearly realize this goal.
> The introduction and experiments mainly read as proposing a new test-time adaptation method and validating its effectiveness, while the interpretability aspect remains secondary. If the intent is to connect mechanistic interpretability with unsupervised adaptation, it would help to explain more clearly how the learned interpretable features affect or explain final model performance, rather than using SAEs and steering vectors mainly as tools.
> In particular, the hypothesis in Appendix A appears central to this connection. I believe the paper would be stronger if this analysis were sufficiently expanded and key points were moved into the main text.
>
> ### **Positioning**
>
> I understand that the current setting (unlabeled and categorical data, discriminative tasks, and ViT-based architectures) is where the method applies most naturally.
> However, if extending the approach to labeled data, continuous outputs, generative tasks, or CNN-based architectures is non-trivial or remains unexplored, the phrase "steering vision foundation models" may overstate the demonstrated scope.
> At present, a more accurate description seems closer to "improving CLIP zero-shot classification."
>
> ### **Fallback Mechanism**
>
> I appreciate that the fallback mechanism was implemented and evaluated.
> My reading of the results, however, is that reconstruction-loss thresholding does **not** currently provide a reliable per-instance fallback criterion.
> If a per-instance criterion is valid, one would expect it to outperform both (1) always using VS2 and (2) never using VS2, if accompanied with appropriate threshold.
> In the reported results, "VS2 (Generalized) + Fallback" does not show this behavior.
> The only slightly favorable cases appear on Flower102 and Caltech101, where the gains over VS2 are marginal (+0.47 vs +0.39, and +1.26 vs +1.19). These results seem not to provide evidence that the proposed fallback mechanism is valid.
>
> ### **Additional VS2++ Baseline**
>
> Regarding the VS2++ setting, my understanding is that it has access to:
>
> - the unlabeled training set,
> - a DINO encoder, and
> - the CLIP zero-shot classifier.
>
> Base on this, my understanding on the kNN baseline is:
> - Extract DINO features for all training samples.
> - For each test sample, retrieve nearest neighbors in DINO feature space.
> - Pass the retrieved training samples through the CLIP classifier.
> - Use majority voting over the resulting pseudo-labels.
>
> If it is right, I wonder if a pseudo-label linear probing baseline could also be considered under the same assumptions:
>
> - Apply the CLIP classifier to all training samples to obtain pseudo-labels.
> - Train a linear classifier on DINO features to predict these pseudo-labels.
> - Use this linear classifier at test time on DINO features.
>
> This would not require labeled data, just like VS2, and could help clarify how much of the VS2++ benefit comes from the proposed mechanism versus simpler pseudo-label based adaptation.

---

> > ### Author Response · Authors · 2025-12-01
> >
> > We would like to thank reviewer **qejr** for the valuable feedback and questions, and we would like to offer the following clarifications:
> >
> > **Additional VS2++ Baseline.**
> >
> > > We thank the reviewer for this helpful suggestion. The **kNN baseline** in VS2++ is implemented exactly as the reviewer described: for each test image, we retrieve neighbors from the unlabeled training set using DINOv2 features, apply the CLIP zero-shot classifier to the retrieved samples, and perform majority voting over the resulting pseudo-labels. In addition, we would like to report the results for the requested **pseudo-label linear probing baseline**. Specifically:
> >
> > > - We apply the CLIP zero-shot classifier to all unlabeled training samples to obtain pseudo-labels.
> > > - We extract frozen DINOv2 features for all training samples.
> > > - We train a multinomial logistic regression linear probe on these features to predict the CLIP pseudo-labels.
> > > - At test time, the linear probe is evaluated on DINOv2 features using ground-truth labels.
> >
> > > The resulting top‑1 accuracies are **56.24%** on CIFAR‑100, **40.78%** on CUB‑200, and **53.52%** on Tiny‑ImageNet. This baseline’s performance gap relative to both CLIP zero-shot and VS2++ might arise from errors introduced during the pseudo-label generation step, i.e., the linear probe is trained on imperfect targets.
> >
> > **Positioning.**
> >
> > > We thank the reviewer for this helpful clarification regarding scope. In the revised version, we aim to further emphasize our setting as operating on a \emph{CLIP-based model}, and specifically on its frozen vision encoder, in the \emph{zero-shot classification} regime. We agree that the phrase “steering vision foundation models” may suggest a broader range of architectures and tasks (e.g., generative models, continuous outputs, CNN backbones), and we are willing to soften this wording. Therefore, we propose the term `“steering visual transformers”` as a more accurate description.
> >
> > > We also understand the suggestion to use the term `“CLIP zero-shot classification.”` We originally used the term `“unsupervised adaptation”` to emphasize that our method leverages unlabeled data from the target domain alongside CLIP’s zero-shot predictions. The standard zero-shot setting typically assumes no access to additional unlabeled data beyond the test instance itself. However, if we adopt the assumption of a pretrained SAE, we acknowledge that the term ‘CLIP zero-shot classification’ may be more appropriate. We are open to adopting this terminology, subject to `consensus` during the discussion phase with reviewers and AC.
> >
> > **Fallback Mechanism.**
> >
> > > We would like to thank the reviewer for their previous comment, which motivated us to implement and evaluate the fallback mechanism. In alignment with Reviewer **fScH**, we aim to reach consensus on its purpose and presentation. We introduced the **Fallback Mechanism** in **Lines 475–485** of the main manuscript and provided additional details in **Appendix O**. We explicitly stated that `“the fallback mechanism primarily serves as a safety measure, defaulting to the baseline on out-of-distribution (OOD) datasets when necessary,”` while also highlighting the development of more nuanced reliability mechanisms as a promising direction for future work.
> >
> > > We also **preserved** the following **sentence in the abstract**: `“Finally, VS2 includes a built-in reliability diagnostic based on SAE reconstruction loss, which is absent in common steering vectors, signaling when steering may underperform and safely triggering a fallback to the baseline.”`
> >
> > **Presentation.**
> >
> > > We would like to thank the reviewer for this comment. We would like to clarify that the primary contribution of VS2 lies in `improving downstream classification performance`, while building upon a body of work that has already explored the interpretability of Sparse Autoencoders (SAEs), such as PatchSAE. Given these prior efforts, we assume the inherent interpretability of SAE features and focus our contribution on demonstrating SAE’s ability to improve classification through targeted steering.
> >
> > > We agree that expanding the **interpretability analysis** would strengthen the paper, and, in response, we have **extended our study to include Class-Conditional Feature Activation analysis** as shown in `Lines 915-948 and Table 7`, where we examine which classes most frequently activate specific SAE features. The changes in the main manuscript have been highlighted in blue. We find that Feature 511 strongly aligns with gull-like categories, while Feature 3067 captures a cross-class visual attribute (e.g., a shared head-and-throat pattern), demonstrating that sparse features can reflect both taxonomic and attribute-level structure

---

> > > ### Author Response · Authors · 2025-12-04
> > >
> > > We would like to thank you for your feedback. If you have any further concerns or questions you would like us to address or clarify, please feel free to let us know.

---

### Official Review · Reviewer_7gbX · 2025-11-01

**Soundness:** 3
**Presentation:** 3
**Contribution:** 2
**Rating:** 4
**Confidence:** 4

**Summary:**

This paper introduces Visual Sparse Steering (VS2), a test-time method that improves zero-shot image classification by steering vision models using sparse features from an autoencoder trained on internal activations—no labeled data required. It amplifies the most salient sparse features identified by the autoencoder to shift embeddings in a meaningful direction, boosting accuracy on CIFAR-100, CUB-200, and Tiny-ImageNet. VS2++ extends this by using unlabeled external images to selectively amplify only the most discriminative features via retrieval, achieving much larger gains when high-quality neighbors are available. Crucially, VS2 includes a built-in reliability check: if the autoencoder’s reconstruction error is high, it falls back to the original model, avoiding harmful steering—a feature absent in prior methods.

**Strengths:**

1. VS2 introduces a novel, label-free test-time steering method for vision models using sparse autoencoder (SAE) features, achieving consistent improvements over CLIP zero-shot.
2. VS2++ extends this with retrieval-augmented selective amplification, demonstrating substantial accuracy gains (up to 21.44% on CIFAR-100) under oracle conditions.
3. Empirical results show robust improvements across diverse datasets (CIFAR-100, CUB-200, Tiny-ImageNet) and ViT backbones.
4. The paper provides thorough ablation studies, hyperparameter sensitivity analyses, and qualitative evidence that SAE latent features capture meaningful visual concepts.

**Weaknesses:**

1. VS2++'s gains (up to 21.44%) rely on oracle positive/negative sets, which are unrealistic in practice; performance drops substantially with noisy pseudo-labels, revealing a critical dependency on high-quality retrieval.
2. The method assumes a **well-trained, in-domain** Sparse Autoencoder (SAE) is available, limiting real-world applicability.
3. While the paper shows sparse features capture semantically meaningful attributes, there's no clear evidence that these features are helpful for classification, i.e., the SAE's reconstruction objective may not inherently capture discriminative visual concepts.
4. The reliance on DINOv2 for retrieval in VS2++ introduces an external, non-trainable component that adds computational overhead and potential bias.

**Questions:**

See the weakness section.

---

> ### Author Response · Authors · 2025-11-25
>
> We thank reviewer **7gbX** for their review. We are pleased that you found our method `“novel,”` with `“empirical results [that] show robust improvements across diverse datasets (CIFAR-100, CUB-200, Tiny-ImageNet) and ViT backbones.”` We appreciate your comments and would like to offer additional insights and clarifications.
>
> ### On In-Domain Sparse Autoencoders and Their Real-World Applicability
>
> > We begin by outlining the main purpose of our paper, followed by key insights into the real-world applicability of our method.
>
> > **Purpose.** The primary goal of our work is to demonstrate to the *mechanistic interpretability* community that **Sparse Autoencoders (SAEs) can be leveraged not only for interpretability but also to improve downstream classification performance**. Most prior works have focused primarily on the interpretability aspect of SAEs. In contrast, we consider it *essential* to demonstrate their **practical utility in downstream tasks**, a direction that remains relatively underexplored.
>
> > **Initial Hypothesis and Experiments.** In this work, we aim to provide a systematic analysis that offers insights into each specific component. Our first question is *whether it is even feasible to steer the model using a Sparse Autoencoder* to improve downstream classification accuracy. To this end, we benchmark our approach using SAEs trained on in-domain data. This ensures that the SAE is sufficiently well-trained to test whether the steering vector improves downstream image classification. We find that it consistently provides performance gains across all tested datasets and ViT backbones. To further strengthen our case, we **include results on six additional datasets** using the same setting. Specifically, we use Aircraft [1], Food101 [2], Flower102 [3], Caltech101 [4], SUN [5], and EuroSAT [6]. For zero-shot evaluation, we use the standard single‑template CLIP prompt (“a photo of a {label}”) for the Baseline, and the 80 ImageNet-style templates introduced by the CLIP authors for the Ensemble setting [7].
>
> | Method / Dataset | Aircraft | Food101 | Flower102 | Caltech101 | SUN | EuroSAT |
> | --- | --- | --- | --- | --- | --- | --- |
> | **Baseline** | 24.84 | 80.71 | 63.72 | 78.53 | 51.79 | 31.61 |
> | **VS2** | 27.69 | 81.20 | 64.81 | 80.95 | 54.80 | 34.65 |
> | **VS2 Gain** | **+2.85** | **+0.49** | **+1.09** | **+2.42** | **+3.01** | **+3.04** |
> |  |  |  |  |  |  |  |
> | **Baseline + Ensemble [7]** | 26.55 | 81.90 | 63.33 | 77.32 | 57.79 | 33.98 |
> | **VS2 + Ensemble [7]** | 28.80 | 82.06 | 63.77 | 80.43 | 60.46 | 36.80 |
> | **VS2+Ens Gain** | **+2.25** | **+0.16** | **+0.44** | **+3.11** | **+2.67** | **+2.82** |
>
> > **Outcome of Initial Experiments.** These experiments on 6 additional datasets **further support our claim that, given a well-trained SAE, it is indeed possible to improve downstream image classification accuracy**. This highlights the importance of effectively scaling and training Sparse Autoencoders, a promising line of research recently emphasized by Gao et al. [8].
>
> > **Towards a More General SAE.** We next aimed to train a more general SAE by incorporating all training datasets during training and evaluating its utility on downstream tasks. While the resulting SAE remains in-domain, we observed that although it maintains strong performance on CIFAR-100 and Tiny-ImageNet, it leads to degraded performance on CUB. Interestingly, we observed that the FVU exceeds 1 on the CUB dataset, underscoring the challenges of training effective SAEs under our current experimental setup. At the same time, it reveals an interesting insight: *reconstruction error may serve as a useful signal for determining a priori when steering is likely to be destructive*. It is important to clarify that FVU is computed across a dataset as an aggregate metric, requiring the mean squared reconstruction error and the variance of the original data. While FVU can be used meaningfully with batched test samples (given sufficient batch size), in this section we shift focus to per-instance evaluation.

---

> > ### Author Response · Authors · 2025-11-25
> >
> > > **Out-of-Distribution Steering with a Generalized SAE.** Specifically, We use the “generalized SAE” of Table 3 trained on CIFAR-100, CUB-200 and Tiny Imagenet and apply steering to the six new datasets that the SAE has not seen during training. We expect the generalized dataset not to necessarily work for all datasets since they will be out-of-distribution and fail for particular ones. We also introduce the **Fallback Mechanism** that is expected to mitigate harmful steering. The fallback mechanism needs one hyperparameter to tune i.e. a *reconstruction threshold*. Essentially when a test sample passes through the network we can calculate using the SAE the reconstruction loss. In the following experiment if the reconstruction loss is below a threshold we apply steering, otherwise we use the original model. We report the following results:
> >
> > |  | Threshold | Aircraft | Food101 | Flower102 | Caltech101 | Sun | EuroSAT |
> > | --- | --- | --- | --- | --- | --- | --- | --- |
> > | **Baseline** | — | 24.84 | 80.71 | 63.72 | 78.53 | 51.79 | 31.61 |
> > | **VS2 (In-domain SAE)** | — | 27.69 (**+2.85**) | 81.20 (**+0.49**) | 64.81 (**+1.09**) | 80.95 (**+2.42**) | 54.80 (**+3.01**) | 34.65 (**+3.04**) |
> > | **VS2 (Generalized SAE)** | — | 18.57 (**−6.27**) | 81.06 (**+0.35**) | 64.11 (**+0.39**) | 79.72 (**+1.19**) | 45.78 (**−6.01**) | 30.76 (**−0.85**) |
> > | **VS2 (Generalized) + Per-Instance Fallback** | **1.5** | 24.87 (**+0.03**) | 80.72 (**+0.01**) | 63.70 (**−0.02**) | 78.53 (**+0.00**) | 51.79 (**+0.00**) | 31.61 (**+0.00**) |
> > |  | **2.0** | 24.87 (**+0.03**) | 80.72 (**+0.01**) | 63.70 (**−0.02**) | 78.53 (**+0.00**) | 51.79 (**+0.00**) | 31.61 (**+0.00**) |
> > |  | **2.5** | 24.87 (**+0.03**) | 80.72 (**+0.01**) | 63.70 (**−0.02**) | 78.53 (**+0.00**) | 51.79 (**+0.00**) | 31.61 (**+0.00**) |
> > |  | **3.0** | 24.87 (**+0.03**) | 80.72 (**+0.01**) | 63.70 (**−0.02**) | 78.53 (**+0.00**) | 51.79 (**+0.00**) | 31.61 (**+0.00**) |
> > |  | **3.5** | 24.87 (**+0.03**) | 80.72 (**+0.01**) | 63.70 (**−0.02**) | 78.51 (**−0.02**) | 51.79 (**+0.00**) | 30.11 (**−1.50**) |
> > |  | **4.0** | 24.87 (**+0.03**) | 80.71 (**+0.00**) | 63.88 (**+0.16**) | 78.11 (**−0.42**) | 51.79 (**+0.00**) | 30.41 (**−1.20**) |
> > |  | **4.5** | 24.87 (**+0.03**) | 80.69 (**−0.02**) | 64.19 (**+0.47**) | 78.42 (**−0.11**) | 51.46 (**−0.33**) | 30.76 (**−0.85**) |
> > |  | **5.0** | 24.12 (**−0.72**) | 80.81 (**+0.10**) | 64.16 (**+0.44**) | 79.39 (**+0.86**) | 49.11 (**−2.68**) | 30.76 (**−0.85**) |
> > |  | **5.5** | 21.81 (**−3.03**) | 81.03 (**+0.32**) | 64.08 (**+0.36**) | 79.79 (**+1.26**) | 46.82 (**−4.97**) | 30.76 (**−0.85**) |
> > |  | **6.0** | 18.93 (**−5.91**) | 81.05 (**+0.34**) | 64.10 (**+0.38**) | 79.70 (**+1.17**) | 46.10 (**−5.69**) | 30.76 (**−0.85**) |
> > |  | **6.5** | 18.69 (**−6.15**) | 81.05 (**+0.34**) | 64.11 (**+0.39**) | 79.72 (**+1.19**) | 45.86 (**−5.93**) | 30.76 (**−0.85**) |
> > |  | **7.0** | 18.57 (**−6.27**) | 81.06 (**+0.35**) | 64.11 (**+0.39**) | 79.72 (**+1.19**) | 45.79 (**−6.00**) | 30.76 (**−0.85**) |
> >
> >
> > > **Outcome of Generalized SAE.** Here, we observe that the “Generalized SAE” yields accuracy gains on Food101, Flower102, and Caltech101, but leads to considerable drops on SUN and Aircraft. **These losses are effectively mitigated by the per-instance Fallback Mechanism**, underscoring its importance for **robust performance in practical applications**.
> >
> > [1] Maji et al. Fine-Grained Visual Classification of Aircraft. 2013.
> >
> > [2] Bossard et al. Food-101 - Mining Discriminative Components with Random Forests. 2014.
> >
> > [3] Nilsback et al. Automated Flower Classification over a Large Number of Classes. 2008.
> >
> > [4] Fei-Fei et al. Learning Generative Visual Models from Few Training Examples: An Incremental Bayesian Approach Tested on 101 Object Categories. 2004.
> >
> > [5] Xiao et al. SUN Database: Large-Scale Scene Recognition from Abbey to Zoo. 2010.
> >
> > [6] Helber et al. EuroSAT: A Novel Dataset and Deep Learning Benchmark for Land Use and Land Cover Classification. 2017.
> >
> > [7] Radford et al. Learning Transferable Visual Models from Natural Language Supervision. 2021.
> >
> > [8] Gao et al. Scaling and Evaluating Sparse Autoencoders. ICLR 2025.

---

> > > ### Author Response · Authors · 2025-11-25
> > >
> > > ### On Discriminative Visual Concepts Emergent from SAE Reconstructions
> > >
> > > > As you correctly noted in Figure 3 of the Appendix, our paper `“shows sparse features capture semantically meaningful attributes”`. These are the same features that are amplified to construct the sparse steering vector, which in turn enhances downstream classification performance. However, we recognize an insightful point in your question regarding the SAE's reconstruction objective: it may not inherently capture discriminative visual concepts. This is a very interesting point raised.
> > >
> > > > In this work, beyond our primary contributions, we also aim to raise awareness within the community about the importance of **task relevance** when training Sparse Autoencoders. Specifically, we distinguish between two tasks: (1) image reconstruction and (2) image classification. While SAEs are typically trained to minimize reconstruction error, **the features learned for this objective may not necessarily align with those that are most discriminative for classification**. Recognizing and addressing this discrepancy is critical for effectively leveraging SAEs in downstream tasks. This is why we introduce PASS in Appendix K, where prototype-based guidance encourages the SAE to extract features that not only support reconstruction but are also biased toward concepts useful for downstream classification.
> > >
> > > > **Similarly, human visual interpretability can be viewed as a distinct task from image reconstruction**. **Features optimized for reconstruction may not align with those that humans find interpretable**. This misalignment might be connected with the phenomena observed in the language domain, such as Feature Splitting and Feature Absorption, as explored by Chanin et al. [9].
> > >
> > > [9] Chanin et al. A is for Absorption: Studying Feature Splitting and Absorption in Sparse Autoencoders. 2025.
> > >
> > > ### On the Oracle Setting of VS2++ and Its Reliance on DINOv2
> > >
> > > > Regarding the performance of VS2++ under the oracle setting, **our goal was to illustrate the potential upper bound of the approach and to motivate future work on more effective feature selection strategies**.
> > >
> > > > In our current setting, the feature selection strategy relies on an RAG-based approach, which introduces the typical computational overhead associated with retrieval-augmented systems, as well as potential biases from the retriever itself. The quality of the retrieved images depends on the quality of the retriever’s embeddings; the better the embeddings, the more relevant and semantically aligned the retrieved examples. We acknowledge this as a valid point; however, we believe that more effective feature selection strategies remain to be explored, leaving ample room for improvement.

---

> > > > ### Author Response · Authors · 2025-12-04
> > > >
> > > > We would like to thank you for your feedback. If you have any further concerns or questions you would like us to address or clarify, please feel free to let us know.

---

### Official Review · Reviewer_Yokp · 2025-11-01

**Soundness:** 3
**Presentation:** 3
**Contribution:** 3
**Rating:** 6
**Confidence:** 4

**Summary:**

This paper introduces Visual Sparse Steering (VS²), a test-time adaptation method for vision foundation models that constructs steering vectors from sparse features extracted by Sparse Autoencoders (SAEs) trained on internal activations. The core insight is that SAEs can identify salient, non-redundant features that, when amplified, improve zero-shot classification without requiring labeled contrastive examples. The authors propose two variants: VS² operates label-free by uniformly amplifying sparse features, while VS2++ selectively weights features using pseudo-labeled neighbors from an external corpus. Experiments on CIFAR-100, CUB-200, and Tiny-ImageNet demonstrate consistent but modest gains (1-4%) over CLIP baselines with VS², and substantial improvements (up to 21.44%) with VS2++ under oracle conditions. The paper also introduces a reliability diagnostic based on SAE reconstruction loss to detect when steering may be harmful.

**Strengths:**

1. The paper addresses an important gap by extending steering vectors to the vision domain without requiring explicit positive/negative examples, which is non-trivial given the entanglement and redundancy of visual representations compared to language tokens.
2. Unlike conventional steering methods, VS² provides a natural fallback mechanism through SAE reconstruction error (Table 3), which signals when test inputs are out-of-distribution and steering should be avoided. This is a practically valuable contribution for deployment scenarios.
3. The sensitivity analyses, ablations on expansion factor and sparsity (Table 7), and per-class accuracy breakdowns (Table 2) provide good evidence that the method is robust to architectural choices and reveals which categories benefit most from sparse steering.
4. The oracle experiments with VS²⁺⁺ (21.44% gains on CIFAR-100) effectively demonstrate the potential ceiling of the approach and motivate future work on better feature selection mechanisms.
5. The paper honestly discusses the reconstruction-vs-alignment tension (Appendix K, Table 10) and acknowledges that features learned for autoencoding are not necessarily optimal for classification, particularly on fine-grained datasets.

**Weaknesses:**

1. The paper primarily compares against CLIP zero-shot and reconstruction baselines but lacks comparisons with other test-time adaptation methods (e.g., TPT, DiffTPT, or prompt-based adaptation techniques). The single comparison with SpLiCE (Table 8) is helpful but insufficient to position the work within the broader test-time adaptation literature.
2. While VS² shows consistent improvements on CIFAR-100 (3-4%), the gains on CUB-200 (0.93-1.08%) and Tiny-ImageNet (1.5-1.84%) are marginal. The paper hypothesizes this stems from features learned for reconstruction not aligning with classification needs, but does not provide sufficient analysis or solutions beyond the prototype-based approach in Appendix K, which requires labels during SAE training.
3. The paper states that VS² is "lightweight" but provides no runtime or memory comparisons. Training SAEs requires processing all layers across entire datasets, and the inference-time reconstruction steps add overhead. Quantitative efficiency analysis relative to the zero-shot baseline would strengthen practical applicability claims.
4. While the geometric intuition is clear (amplify sparse features -> steer toward salient directions), the paper lacks formal analysis of why autoencoding objectives should discover classification-relevant features, or under what conditions this alignment holds. The empirical observation that it works on CIFAR-100 but struggles on CUB-200 suggests important gaps in understanding.
5. Evaluation scope limitations: All experiments use CLIP ViT backbones; generalization to other architectures (ConvNets, other VLMs) is unexplored. Only classification tasks are evaluated; applicability to detection, segmentation, or other vision tasks remains unclear. The VS²⁺⁺ non-oracle results (Table 1) often show smaller gains than standard steering vectors, undermining claims about SAE superiority when supervision is noisy
6. The term "contrastive data" is used loosely (the method doesn't require labeled positives/negatives but still uses contrastive loss in training). Section 3.2.2 could better explain how pseudo-labeling quality affects VS²⁺⁺ performance and why negative groups from top-N neighbors constitute "hard" cases.

**Questions:**

1. Can you provide wall-clock time and memory comparisons between VS², standard CLIP inference, and other test-time adaptation methods? How does SAE training time scale with dataset size and model depth?
2. Table 12(b) and 12(d) show classes where VS² hurts performance. Can you characterize these failure modes? Are there systematic patterns (e.g., texture-based vs. shape-based classes, frequency in training data) that predict when steering will be harmful beyond reconstruction error?
3. Have you tested VS² with other vision encoders (DINOv2, MAE, supervised ImageNet models)? Does the method's effectiveness depend on CLIP's specific training objective or architecture?
4. How does VS² compare with learnable prompt methods (CoOp, CoCoOp) or adapter-based approaches that also aim to improve zero-shot performance? These methods require some labeled data but might provide more principled comparisons than pure zero-shot.
5. The prototype-alignment approach (PASS) in Appendix K shows promise but requires labels. Could a clustering-based approach discover multiple prototypes per class in an unsupervised manner? Have you explored hierarchical or mixture models for class representations?

---

> ### Author Response · Authors · 2025-11-25
>
> We would like to thank reviewer **Yokp** for the detailed review and the insightful comments. We are pleased that you noted our paper `“addresses an important gap by extending steering vectors to the vision domain without requiring explicit positive/negative examples, which is non-trivial,”` and recognized the `"natural fallback mechanism"` as `“a valuable contribution for deployment scenarios.”` We truly appreciate your input, and we would like to provide further insights.
>
> ### Regarding Comparisons with other Baselines
>
> > We would first like to (1) clarify the rationale and motivation behind our experimental setup, and (2) provide additional insights by incorporating stronger baselines from the Test-Time Adaptation (TTA) literature, further positioning our paper within this domain.
>
> > The main purpose of our work is to showcase to the *mechanistic interpretability* community that **Sparse Autoencoders (SAEs) can be leveraged not only for interpretability, but also to improve downstream classification performance**. This is an underexplored direction within the community, and our intention was to position the paper accordingly. That said, we do appreciate your insight regarding *comparisons and positioning with respect to the Unsupervised Domain Adaptation (UDA) literature*. While not a contribution of our work, an *important distinction* of our approach is that *VS2 is inherently interpretable*, a property we think is valuable. We provide supporting evidence for this in Appendix A.2.
>
> > While interpretability is not a focus of most existing UDA or Test-Time Adaptation (TTA) literature, **we agree that providing comparisons with representative TTA methods is essential**. Accordingly, for the datasets in Table 1 of our main paper, we include baselines from TPT [1] , C-TPT [2], and Diff-TPT [3]. These methods optimize text prompts during adaptation, whereas our method does not modify or update the text prompts. We focus our comparsions for these Test-Time Adaptation methods since CoOP [4] and CoCoOp [5] require access to labeled data for training, and thus they are not directly comparable to our *fully unsupervised method*, which operates without any labeled supervision. In our Baseline setting, we use the fixed prompt “a photo of a {label}.” The Ensemble setting corresponds to a set of 80 fixed ImageNet-style templates, following CLIP’s zero-shot evaluation setup [6]. Finally, Aug refers to the test-time image augmentations used in TTA baselines. For efficiency, we apply only 10 image augmentations, compared to the 63 used in other methods.
>
> |  | CIFAR-100 | CUB-200 | Tiny-Imagenet |
> | --- | --- | --- | --- |
> | Baseline | 61.07 | 51.76 | 56.64 |
> | Ensemble [4] | 63.66 | 51.54 | 61.39 |
> | TPT [1]  | 64.09 | 51.83 | 62.77 |
> | C-TPT [2] | `64.86` | 52.54 | **`63.20`** |
> | Diff-TPT [3] | 63.04 | **`52.80`** | 61.60 |
> | VS2 (ours) | 64.52 | `52.69` | 58.14 |
> | VS2 (ours) + Ensemble [4] | **`65.48`** | 52.47 | `62.79` |
> | VS2 (ours) + Ensemble [4] + Aug | 65.80 | 51.54 | 62.59 |
>
> > We observe that our method *performs comparably* to existing Test-Time Adaptation techniques, achieving the **best accuracy on CIFAR-100** and **second-best performance on CUB-200 and Tiny ImageNet**. However, a key *advantage of our approach* is its **inherent interpretability**, an aspect not present in the compared methods. Moreover, our method is potentially modality-agnostic and could be extended to both language and vision domains.
>
> [1] Shu et al. Test-Time Prompt Tuning for Zero-Shot Generalization in Vision-Language Models. 2022.
>
> [2] Yoon et al. C-TPT Calibrated Test-Time Prompt Tuning for Vision-Language Models via Text Feature Dispersion. 2024.
>
> [3] Feng et al. Diverse Data Augmentation with Diffusions for Effective Test-Time Prompt Tuning. 2023.
>
> [4] Zhou et al. Conditional Prompt Learning for Vision-Language Models. 2022.
>
> [5] Zhou et al. Learning to Prompt for Vision-Language Models. 2022.
>
> [6] Radford et al. Learning Transferable Visual Models from Natural Language Supervision. 2021.

---

> > ### Author Response · Authors · 2025-11-25
> >
> > ### Wall-clock time, Memory Comparisons, and Scaling Laws
> >
> > > To clarify the **computational overhead** of our method, we report the exact <u>inference</u> **FLOPs**, **MACs**, and **parameter counts** for the CLIP ViT-B/32 vision encoder under the baseline model, VS2, and LoRA.
> >
> >
> > | Method | GFLOPs (fwd) | GMACs (fwd) | Params (M) |
> > | --- | --- | --- | --- |
> > | Plain CLIP (vision) | **8.7295** | **4.3623** | **87.456** |
> > | VS2 (SAE steering) | **8.7342** | **4.3647** | **92.178** |
> > | LoRA (rank = 16) | **8.7885** | **4.3918** | **88.046** |
> >
> > > VS2 **adds only 0.0047 GFLOPs over the baseline**, less than 0.1%, because it applies a lightweight SAE operation to the *CLS token in a single layer*, whereas LoRA introduces additional projections inside every attention block across all tokens and layers, leading to noticeably higher FLOPs despite having fewer parameters.
> >
> > > To further quantify the cost of adapting CLIP, we report per-sample <u>training</u> FLOPs and trainable parameters for SAE (with and without cached activations), LoRA, and full fine-tuning on the ViT-B/32 vision encoder.
> >
> > | Method | Training FLOPs (fwd + bwd) | FLOPs (G) | Trainable Params |
> > | --- | --- | --- | --- |
> > | SAE (cached activations) | 14,164,992 | **0.014** | 4,722,432 |
> > | SAE (non-cached; incl. extraction) | 8,743,645,440 | **8.744** | 4,722,432 |
> > | LoRA (r = 16) | 26,365,388,544 | **26.365** | 589,824 |
> > | Full Fine-tuning (vision encoder) | 26,188,441,344 | **26.188** | 87,456,000 |
> >
> > > Even when including activation extraction, SAE remains over around 3× cheaper than LoRA or full fine-tuning, and when cached activations are used, as in VS2, the training cost drops to only 0.014 GFLOPs per sample.
> >
> > > Finally, **Test-Time Adaptation (TTA) methods**, such as TPT and its variants, introduce **substantial computational overhead**. This is primarily due to the use of *multiple image augmentations*, *repeated test-time optimization steps*, and the *requirement to backpropagate gradients to update prompts during inference*. A plain CLIP ViT-B/32 forward pass on a single 224×224 image, together with 100 text prompts (one for each class given that the number of labels is 100), costs 590.6 GFLOPs in total (8.7 GFLOPs for the vision encoder and 581.8 GFLOPs for the text encoder). In contrast, in our current testbed, TPT performs test-time adaptation by generating 63 augmented views of the test image and running 4 optimization steps, each of which requires a full CLIP forward pass over all views plus a backward pass through the text encoder. This results in 9.217 TFLOPs of adaptation cost per test image, followed by a final CLIP inference of 0.591 TFLOPs, for a total of 9.807 TFLOPs. Overall, with these computations, TPT requires 16.6× more computation per image than plain CLIP. Importantly, this overhead is incurred at inference time, making TPT far more expensive than lightweight steering methods such as VS2, which increase per-image inference cost by less than 0.01%.
> >
> > > **Regarding SAE size**: The SAE consists of a learned encoder $E \in \mathbb{R}^{768 \times 3072} $ and decoder $D \in \mathbb{R}^{3072 \times 768}$. We provide details in Appendix C. The expansion factor of 4 indicates that the sparse feature dimensionality is 4× larger than the original embedding size, enabling a richer sparse representation while maintaining reasonable inference overhead. Notably, the by-definition sparsity (e.g., using top-64 active features with the rest set to zero) also helps minimize computational overhead during decoding. Specifically, when the sparse latent vector \( z \) is multiplied with the decoder \( D \), we only need to compute the contributions from the 64 non-zero features, avoiding unnecessary multiplications with zeros.
> >
> > > Regarding scaling, model depth should not pose a significant issue in our setup, as we only require a single forward pass through the model to store the necessary representations for training the SAE. Regarding the *scaling behavior* of the Top‑k Sparse Autoencoders used in our work, we refer the reviewer to OpenAI’s recent study on Top‑k SAEs [7], which provides detailed analysis and empirical scaling laws for this class of models.
> >
> > [7] Gao et al. Scaling and Evaluating Sparse Autoencoders. ICLR 2025.

---

> > > ### Author Response · Authors · 2025-11-25
> > >
> > > ### Rationale Behind Using CLIP
> > >
> > > > We use CLIP as our foundation because it enables evaluation in a fully unlabeled setting, which aligns with our focus on unsupervised adaptation. While our main focus is on downstream classification, the broader methodology is not limited to Vision Transformers or [CLS] token representations. For other architectures, including those without [CLS] tokens or even closed-source models, one could train SAEs on the final pooled embeddings produced by the model. Beyond classification, similar ideas have begun to emerge in related domains: for example, SAEs have been applied to concept unlearning in diffusion models [8] and multimodal alignment in large language–vision models [9].
> > >
> > > [8] Cywiński et al. SAeUron Interpretable Concept Unlearning in Diffusion Models with Sparse Autoencoders. 2025.
> > >
> > > [9] Lou et al. SAE-V Interpreting Multimodal Models for Enhanced Alignment. 2025.
> > >
> > > ### Failure Modes of Visual Sparse Steering
> > >
> > > > This is a very interesting and challenging question. In this particular case, however, we do not believe that class frequency is the underlying cause of failure, as the SAE training data for CIFAR-100 is relatively balanced, containing 50,000 images evenly distributed across 100 classes (i.e., 500 images per class). This provides a dense and uniform training signal, suggesting that other factors, such as probably feature complexity or inter-class similarity, may be responsible.
> > >
> > > ### On the Prototype-Alignment approach (PASS)
> > >
> > > > We are pleased that you reviewed our method PASS in the appendix and appreciate your insightful suggestions. We included PASS in the Appendix because it requires access to labels, which violates the fully unsupervised setting that defines the core contribution of this work. Through this work, we came to understand and hope to communicate to the community that reconstruction objectives can lead to features that are not necessarily relevant for downstream classification tasks. As a side note, through the rebuttal process, we have begun to see that this insight, regarding the mismatch between reconstruction and classification relevance, may also extend to interpretability. If we consider human-aligned interpretability as its own task, distinct from reconstruction, this opens the door to understanding phenomena like feature absorption through a task-specific lens [10].
> > >
> > > > Returning to the main question, we believe it is a valuable direction to explore multiple prototypes per class in an unsupervised manner. This approach has the potential to capture intra-class variation by modeling different modes within a class, such as cars of different colors, types, or even varying viewpoints of the same object. Finally, exploring hierarchical or mixture models for class representation is an area of active interest for us and a promising direction for future work.
> > >
> > >
> > > [10] Chanin et al. A is for Absorption: Studying Feature Splitting and Absorption in Sparse Autoencoders. 2025.

---

### Author Response · Authors · 2025-12-04
**Final Remarks and Acknowledgments**

We thank the reviewers for their thoughtful feedback and the Chairs for facilitating the discussion. Following the discussion phase, we are pleased to note that `Reviewer fScH stated most concerns were resolved and indicated an intention to raise their score from 4 to 6`. We summarize below how the final version addresses key points:

**Unsupervised Domain Adaptation (UDA), Test-Time Adaptation (TTA) and Baselines**

Reviewers **fScH** and **Yokp** asked us to better situate VS2 relative to UDA and TTA literature:

- We `added a UDA/TTA` paragraph to the `related work`, explicitly positioning VS2 within this literature.
- We `included TPT, C-TPT, and Diff-TPT as baselines` and show that `VS2 performs comparably to existing TTA techniques`, achieving the **best accuracy on CIFAR-100** and **second-best performance on CUB-200 and Tiny ImageNet**, while additionally *offering* inherent `interpretability` and `lower inference-time overhead`. These comparisons are now part of the main paper.

**Extending the Evaluation of VS2 to 6 Additional Datasets**

We extended the evaluation of `VS2 to six additional datasets`, i.e., Aircraft, Food101, Flower102, Caltech101, SUN, and EuroSAT, and observed that `VS2 provides consistent performance gains across all six datasets`, with improvements of up to +3.04%.

**“Lightweight” Claim and Computational Overhead**

Reviewers **Yokp** and **fScH** requested concrete evidence to support the claim that VS2 is a lightweight adaptation method. In response, we:

- `Added detailed FLOPs, MACs, and parameter counts` comparing CLIP, VS2, and LoRA, as well as **per-sample training FLOPs** for SAE, LoRA, and full fine-tuning.
- Quantitatively demonstrated that `VS2 adds minimal inference FLOPs over CLIP`, and that `SAE training, especially with cached activations, is significantly more efficient` than both LoRA and full fine-tuning.

**On the inherent VS2 Fallback Mechanism through Self-Reconstruction**

Reviewers **qejr** and **fScH** inquired about the fallback mechanism’s practical utility. In response, `we implemented and evaluated a per-instance fallback strategy based on SAE reconstruction loss`, applied to out-of-distribution datasets.

On out-of-distribution datasets, a generalized SAE yielded accuracy gains on Food101, Flower102, and Caltech101, but showed notable drops on SUN and Aircraft. These `failures were effectively mitigated by the fallback mechanism`, which dynamically applies steering only when the reconstruction loss is below a threshold.

We clarify in the manuscript that `the fallback serves as a safety mechanism, reverting to baseline predictions when steering is unreliable`, and we highlight future work on more refined reliability diagnostics.

**Clarifying Scope and Terminology: “Steering Visual Transformers” and Unsupervised Adaptation**

In discussion with Reviewer **qejr**, we agreed to adopt the softer phrasing `“steering visual transformers”` to more accurately reflect the current empirical scope of our work. Upon consensus, we are also open to using the term `“CLIP zero-shot classification”`, though we note that our original use of `“unsupervised adaptation”` was intended to emphasize that our method leverages *unlabeled data from the target domain* alongside CLIP’s zero-shot predictions. We remain open to any further suggestions regarding terminology or framing.

**Connection to Mechanistic Interpretability.**

To better realize our stated goal to the mechanistic interpretability community, we:

- Emphasized that the `core contribution` is to show that SAE features, originally studied for interpretability, can be used `to improve downstream classification via steering`. The paper is thus focused on downstream classification improvements. Interpretability is explored in other works, such as PatchSAE.
- We, though, provided evidence of the inherent interpretability in Appendix A2 of the original paper. We also `expanded the interpretability analysis` with class-conditional activation patterns for specific SAE features, demonstrating that they may capture both taxonomic structure and cross-class attributes.

We are pleased to note that, `during the discussion phase, we were able to clarify and further strengthen the paper thanks to the reviewers' thoughtful feedback`. Our work presents the following key contributions:

- A **novel and lightweight unsupervised adaptation method (VS2)** for CLIP-based ViT classifiers, with `performance improvements` across diverse datasets;
- A demonstration that **sparse, inherently interpretable features** learned by Sparse Autoencoders (SAEs) can be used not only for interpretability but also to consistently `improve downstream classification performance`;
- The introduction of a **reliability diagnostic and fallback mechanism** absent in other methods, which serves as a `practical safety measure in out-of-distribution scenarios`, and lays the groundwork for future research.

---

### Meta-Review · Area_Chair_ywYt · 2026-01-07

**Summary:**

This submission was reviewed by four expert reviewers, with the ratings of: 3 borderline reject, and 1 borderline accept. The main concerns from the reviewers are around the lack of comparison with other test-time adaptation methods, marginal performance gains, missing runtime/memory analysis, lack of evidence to support the claims, limited evaluation scope, the contrastive data, practicality, computational overhead and potential bias, overclaimed contributions, contradictory premise for data-scarce scenarios, reliability diagnostic, insufficient related work discussion and comparison. The authors provided a rebuttal for the raised concerns, and the reviewers engaged in the discussions.

After carefully going through all the review comments, the authors' rebuttal, and the discussions, it can be seen that some concerns are addressed by the rebuttal and further experiments from the authors. However, there are still major concerns remaining not well addressed. Although some interesting findings were presented, there is no strong support for a clear acceptance while major concerns are still left not well addressed. As a result, it is unfortunate that this paper in its current form is not ready for publication in ICLR, and needs a major revision followed by another round of review for assessment. But the authors are encouraged to further improve their paper accordingly and consider submitting to a future venue.

**Reviewer Concerns:**

Concerns that the AC thinks were addressed by the rebuttal: comparison to test-time adaptation methods; more baselines for VS2++; and other minor concerns.

Concerns that are still outstanding: marginal performance gains; runtime and memory analysis;  lack of analysis of autoencoding objectives; evaluation scope limitations; concerns about the contrastive data; the reliance on high-quality retrieval and practicality; evidence for the sparse features being helpful for classification; the overclaims and misleading claim; the contradictory premise; unvalidated reliability diagnostic; related work comparison (e.g. unsupervised domain adaptation).

**Reviewer Scores:**

According to the review comments, and the rebuttal, for each review the reviewer might have changed their score in the way below, if they had been able to participate fully in the discussion:
* Reviewer Yokp: borderline accept to borderline reject or reject
* Reviewer 7gbX: borderline reject to reject, or unchanged
* Reviewer qejr: borderline reject to reject
* Reviewer fScH: borderline reject to borderline accept.

---

### Decision · Program_Chairs · 2026-01-26

Reject